# Performance of Strengthened, Reinforced Concrete Shear Walls with Opening

Hala Mamdouh [1], Nasr Zenhom [1], Mahmoud Hasabo [1], Ahmed Farouk Deifalla [2,*] and Amany Salman [1]

1 Civil Engineering Department, Faculty of Engineering, Helwan University, Cairo 11718, Egypt
2 Structural Engineering and Construction Management Department, Future University in Engineering, Cairo 11835, Egypt
* Correspondence: ahmed.deifalla@fue.edu.eg

**Abstract:** Shear walls are one of the primary lateral resisting structural elements. Due to architectural and technical needs, openings in the structural wall are almost inevitable. Discontinuity regions and a reduction in wall stiffness result from these openings. The use of fiber-reinforced-polymer (FRP) systems is a sustainable construction solution for strengthening these areas and is a viable method to restore their integrity and serviceability. This paper presents an experimental and analytical study on the behavior of reinforced concrete (RC) shear walls with openings of various sizes and positions strengthened using glass-fiber-reinforced-polymer (GFRP) sheets. Ten RC shear walls were cast and tested; initially without strengthening; and then retested with a layer of bi-directional GFRP sheet added around the opening. The finite-element (FE) program ANSYS was used for modeling since using FE contributes to sustainability. The results showed that for un-strengthened walls with a 6.25% opening and strengthened walls with an 11.11% opening, the rate of stiffness degradation was reasonably low. As the opening size was enlarged, the strength and stiffness values were drastically reduced; and the shear walls with an opening at the mid-height position also have smaller load capacities compared to the bottom and top opening positions. In addition, the ability of the GFRP sheets to control stress redistribution and crack propagation improved the overall performance of the walls. The FE and experimental results match well. Furthermore, the ACI and ECP calculations revealed a good prediction of lateral load capacity without considering the opening position, whereas the other proposed models were inaccurate. Finally, the author proposed a reduction factor (β) to the shear strength equation provided by ECP-203-2020 depending on openings sizes and locations; and suggests that FRP sheets be used around openings to assure the appropriate performance and avoid unexpected failure.

**Keywords:** shear wall; opening; strengthening; glass fiber; finite element (FE)

## 1. Introduction

In structural engineering, a shear wall is a vertical element that resists the lateral forces due to their high in-plane rigidity. Openings in the structural wall are almost inevitable due to architectural and technical requirements, and these cut-out openings of shear walls after construction lead to discontinuity zones and a decrease in wall rigidity. A viable method for improving the flexural and shear capacities is to strengthen these areas using fiber-reinforced-polymer (FRP) systems. This paper presents an experimental and analytical study on the behavior of reinforced concrete (RC) shear walls with openings strengthened using glass-fiber-reinforced-polymer (GFRP) sheets.

The shear wall is categorized as a squat wall or slender wall according to its aspect ratio (height to length ratio). In a squat wall, its aspect ratio is less than or equal to 2.0 [1–3]. In the squat shear walls, the shear transfer occurs by the truss action, which provides a stiffer system than that for slender walls [4]. The lateral loads from earthquakes or wind loads and axial loads cause different modes of failure, including diagonal compression,

diagonal tension, or sliding shear [5]. Shear wall openings result in discontinuity regions, geometric load flow constraints, and a reduction in wall rigidity and stiffness. Furthermore, there is a concentration of stress at the corner of these openings.

FRP composites material are one of the most popular options for tackling the aforementioned issues and are commonly applied for strengthening elements [6–10]. At the end of the 1980s and early 1990s, the FRPs composites were applied for structural strengthening, and after that time, this repairing method spread rapidly. The FRPs are characterized by their superior properties, such as their light weight, corrosion resistance, high strength, and ease of installation [11–13]. GFRP sheets are distinguished by their low price compared to the other FRP materials. The FRP sheets externally bonded to the surface of RC members improve the load capacities and seismic resistance. They also do not support or spread fire when heated, do not produce toxic products or smoke, and can be recycled using different methods with minimum effect on nature [14,15].

Ehsani et al. [16] used GFRP to repair the structural walls in a high-rise building after the Northridge earthquake to display the advantages of this repair method. After strengthening, the moment capacity of a unit width wall increased to 74%, whereas before the strengthening, it was 13.8%.

The bond properties between concrete and CFRP under in-plane cyclic quasi-static load were investigated experimentally by Volney et al. and Lombard et al. [17,18]. They identified three types of failure: fiber composite failure, fiber delamination from the concrete surface (cohesive failure), and concrete surface shear failure. Moreover, a theoretical model was developed for predicting the load-displacement curve of RC shear walls strengthened by CFRP.

Hiotakiset al. and Nagy et al. [19,20] studied the behavior of RC shear walls retrofitted with FRPs and subjected to lateral load; different strengthening schemes were used, and it was found that the average failure load increased by (45–132%), and the elastic limit increased by 47%.

Dejian Shen et al. [21] studied the seismic performance of RC shear walls strengthened with basalt-fiber-reinforced-polymer BFRP in various configurations to investigate failure modes, ductility ratio, stiffness characteristic, energy-dissipation capacity, and load-carrying capacity, and the results revealed that the use of BFRP strips significantly improved the seismic performance of RC shear wall under cyclic load.

Mohammed Nagib et al. [22] presented an experimental investigation conducted on seismically shear-deficient squat-reinforced concrete (RC) shear walls strengthened with casting ultra-high-performance fiber-reinforced concrete (UHPFRC) layers/jacket subjected to lateral quasi-static reversed cyclic loading and a constant axial load. According to the results, The UHPFRC-strengthened techniques increased the lateral load-carrying capacities by 70% to 227% compared to the original wall. In addition, results indicated the strengthening techniques significantly improved ductility and energy-dissipation capacities.

A study by Visar et al. [23] dealt with analyzing the structural responses of glass-fiber-reinforced polymer (GFRP) tubes filled with recycled and concrete material for developing composite piles as an alternative to traditional steel-reinforced piles in bridge foundations. The lateral strength of the GFRP composite pile and pre-stressed piles were investigated under both axial compression and bending moment loads. Based on the conducted parametric study, the required axial and bending capacities of piles in different ranges of eccentricities can be reached using the combination of tube wall thickness and GFRP fiber percentages.

Mohsen et al. [24] used a new method named multi-pier (MP), which is fast and accurate to determine the behavior of perforated unreinforced masonry (PURM) walls. The outcomes of the MP method were employed to predict the behavior of PURM walls using various machine learning approaches. Results indicated that the adjacent piers of opening had a remarkable impact on the overall response of the PURM wall. Finally, the ability of the MP method to conduct stochastic analysis was evaluated.

Mohammad and Mansour [25] studied the effect of openings on the behavior of a composite steel plate shear wall CSPSW through the experimental test and finite-element

simulation in ABAQUS and proposed different methods to reduce the negative effect of the opening on the behavior of the system. They found that including an opening to CSPSW reduces the system's stiffness and energy absorption capacity, which increases displacements. However, using a reinforcing steel plate around the opening is more efficient and helps to restore some of the overall stiffness lost.

Naci Cagler et al. [26] conducted an experimental investigation to study the behavior of RC shear walls found in old and existing buildings that do not follow the design rules in recent earthquake standards. Four specimens represent nonconforming shear walls, and one wall is used as a reference specimen designed by recent building codes using deformed bars. The results showed a substantial loss of stiffness, ductility, and energy dissipation for the tested nonconforming shear walls. Furthermore, these specimens revealed bar slip phenomena, which contributed to more than 80% of the total lateral displacement capacity. In contrast, the reference shear wall exhibited a notable flexural behavior and plastic hinge formation. Additionally, the shear walls built with smooth reinforcement bars lost about 44% of their theoretical potential flexural capacity due to the observed bar slip failure.

In this paper, the ANSYS parametric design language (APDL) finite-element program is used to simulate the behavior of GFRP-strengthened RC shear wall under static loading; the FE technique is a cost-effective numerical method for solving difficult physical problems with acceptable approximation [27]. Recently, the analysis of nonlinear fracture mechanics in FE has been developed; the cohesive zone model (CZM) is a simple method that uses a contact element to describe the interface surfaces of two materials [28,29]. Muhammad et al. [30] used ANSYS for modeling the shear wall with varying percentages of base opening. It was reported that the stiffness degradation is quite low, up to 50% base opening. Beyond this limit, the rate of decrease in strength and stiffness is excessive.

El-Kashif et al. [31] developed numerical modeling using the ANSYS program; the effect of using FRP sheets to improve the seismic behavior of RC shear walls was investigated in this study. The result revealed that FRPs successfully eliminate the brittle shear failure in walls, and the numerical outputs in terms of load-displacements agreed well with the experimental data.

Mohamed Husain [32] developed a 3D nonlinear finite-element model on ABAQUS to investigate the seismic response of shear walls with openings strengthened with FRP wraps under monotonic loads. The proposed FE model was validated with data from previous experiments. The FE results showed that the proposed scheme of CFRP laminates significantly enhances the ductility and energy dissipation of the shear wall as well as increasing the lateral load strength and deformation capacity.

As recommended by Doh and Fragomeni [33], "More research on the subject of concrete walls with various openings is still relatively unexplored and will require more focused investigation in the future". Moreover, Mosallam et al. [34] recommended that more rigorous experimental and analytical studies be performed to measure the various factors and shear wall retrofitting schemes.

## 2. Research Significance

Some concrete structures (especially those with openings) require strengthening to restore their integrity and serviceability, so demand for sustainable reinforcement materials is required. GFRP reinforcement is quickly becoming a viable alternative to traditional materials and a sustainable construction solution due to its outstanding properties.

The importance of this research is to investigate experimentally and analytically the behavior of strengthening shear walls with opening to achieve the following objectives:

1.  Examine the effect of different parameters on the behavior of a shear wall with an opening subjected to lateral load synchronized with axial load;
2.  Identify the optimum opening size and location;
3.  Investigate the effect of proposed scheme of GFRPs composites for strengthening RC shear walls with openings;

4. Validate a numerical model to simulate the behavior of the GFRP-strengthened RC shear wall with the opening;

5. Examine the proposed models for squat wall strength and deformation capacity and demonstrate their accuracy.

## 3. Experimental Program

The experimental work was developed in the Civil Engineering Laboratory at Helwan University for testing ten RC shear walls (one solid and nine with opening) subjected to lateral load with a constant axial load of $0.04\,A_gf'_c$; this axial load value was selected based on relevant studies as well as the fact that the presence of axial loads increases shear strength and flexural strength and minimizes sliding shear [35,36]. Then, the tested walls with openings retrofitted and strengthened by GFRP sheets were retested up to failure.

### 3.1. Material

3.1.1. Gravel, Sand, Cement, and Water

Locally available materials (sand, gravel with a maximum size of 15 mm, ordinary Portland cement, and potable water) conforming to ECP 203, 2020 [2] were used in the experimental study. All batches used were of high quality and were clean and free from impurities, as shown in Figure 1.

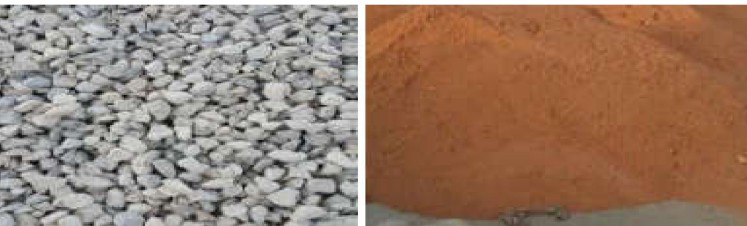

**Figure 1.** Gravel and sand used in experimental work.

3.1.2. Concrete and Reinforcement Steel

Four cubes (150 × 150 × 150 mm) were tested to determine the compressive strength of the concrete. The average compressive strength was about 31 MPa after 28 days of curing. In addition, four cylinders measuring (150 mm × 300 mm) were tested using the Universal Testing Machine to determine the splitting tensile strength of concrete; the average split tensile strength was about 3.34 MPa. Table 1 displays the compressive strength results of tested cubes and the splitting tensile strength results of tested cylinders, and Figure 2 illustrates the testing machines and the typical mode of failure.

**Table 1.** Test results of compressive strength and splitting tensile strength.

| Cube No. | Compressive Strength ($f_{cu}$) MPa | Cylinder No. | Splitting Tensile Strength ($f_{sp}$) MPa |
|---|---|---|---|
| Cube 1 | 31.40 | Cylinder 1 | 3.39 |
| Cube 2 | 33.60 | Cylinder 2 | 3.26 |
| Cube 3 | 29.96 | Cylinder 3 | 3.52 |
| Cube 4 | 29.42 | Cylinder 4 | 3.21 |
| Average $f_{cu}$ | 31.1 | Average $f_{sp}$ | 3.34 |

Table 2 gives the mix design composition per 1 m³ volume of concrete used in the tested shear walls. Two diameters (6 mm and 8 mm) of mild smooth steel were used with 280 MPa yield strength. Mild steel bars were smooth and without ribs. They were used in this study because of their flexibility, which allows for easy cutting and bending without damage.

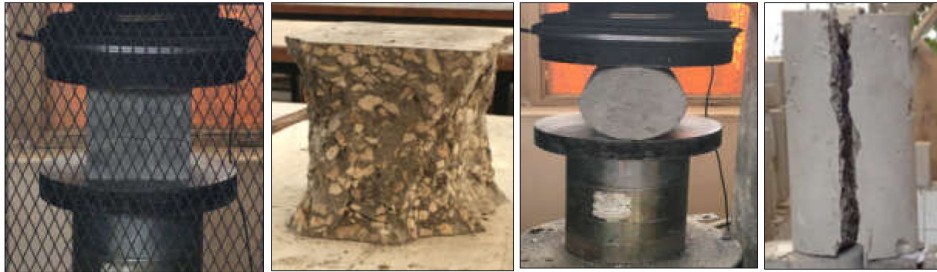

**Figure 2.** Concrete compressive strength test and splitting tensile strength test.

**Table 2.** Mix Design for 31 MPa grade concrete.

| Material | Weight (kg) |
|---|---|
| Cement (kg) | 350 |
| Coarse aggregate (kg) | 1320 |
| Fine aggregate (kg) | 640 |
| Water (kg) | 150 |
| Water/cement ratio (w/c%) | 0.42% |

### 3.1.3. GFRP Sheets and Resin

GFRP was chosen for this study due to its availability and low cost compared to the other FRP types. GFRP is becoming a sustainable construction solution due to its outstanding properties. GFRP has resistance to chemicals, environment, heat, and salt. It has a high strength-to-weight ratio. Furthermore, GFRP is considered a distinguished insulation and low-cost strengthening material [37]. The used GFRPs sheet (SikaWarp-430 G) was a bi-directionally woven, designed for installation using the dry or wet application process, as shown in Figure 3; the properties of glass fiber sheets and epoxy resin are listed in Tables 3 and 4, respectively. The resin was mixed in accordance with the FRP system manufacturer's recommended procedure.

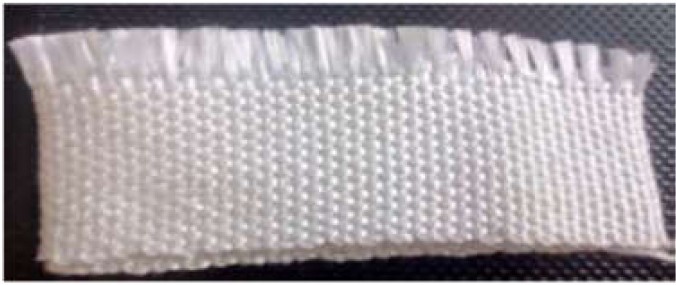

**Figure 3.** The utilized glass fiber sheets.

**Table 3.** Properties of glass fiber sheets (SikaWarp-430 G).

| | |
|---|---|
| Dry Fiber Density | 2.56 g/cm$^2$ |
| Area density | 430 g/cm$^2$ |
| Dry fiber tensile strength | 2500 N/mm$^2$ |
| Dry modulus of elasticity in tension | 72,000 N/mm$^2$ |
| Dry fiber thickness | 0.168 mm |

**Table 4.** Properties of Epoxy (Sikadur-330).

| Density | $1.30 \pm 0.1$ kg/L |
|---|---|
| Modulus of elasticity in flexural | 3800 N/mm$^2$ |
| Modulus of elasticity in tension | 4545 N/mm$^2$ |
| Tensile strength | 30 N/mm$^2$ |
| Elongation at break | 0.66% |

### 3.2. Specimens Details and Dimensions

Table 5 shows the notations used for the names of specimens and groups in the current investigation, which includes 10 specimens before applying glass fiber sheets categorized into four groups. The test specimens were about one-third the size of the shear walls utilized in a multistory structure.

**Table 5.** Notations for the Names of Specimens.

| SW | Shear Wall |
|---|---|
| L | Height of the wall = 750 mm |
| W | Width of the wall = 750 mm |
| N | No opening |
| B | The location of opening at the bottom |
| M | The location of opening at the middle |
| T | The location of opening at the top |

All RC shear walls were cast with the same dimensions of 750 mm width, 750 mm height ($h_w/l_w = 1$), and 70 mm thickness, monolithically connected to an upper and lower beam. These dimensions were tested by Lefas et al. [38]; the arrangement of opening and the scheme of the applied GFRPs were proposed by Behfarnia et al. [39]. The upper beam dimensions (1150 mm × 200 mm × 150 mm) functioned as the element through which axial and horizontal loads were applied to the walls and as a cage for the anchorage of the vertical bars. The lower beam dimension (1150 mm × 200 mm × 300 mm) was utilized to clamp the specimens down to the laboratory floor. Concrete dimensions, the location of hydraulic jacks, and the reinforcement details are shown in Figure 4. All specimens were divided into four main groups based on the opening location. The first group N has no opening, the second group B has openings at the bottom, the third group M has openings in the middle, and the fourth group T has openings at the top. Square-shaped openings were installed and varied in size (L/4, L/3, and L/2) (L corresponded to the length of the wall). Table 6 and Figure 5 illustrate the geometry of the shear wall, arrangements of openings, and the dimensions of the applied GFRP sheet around the opening.

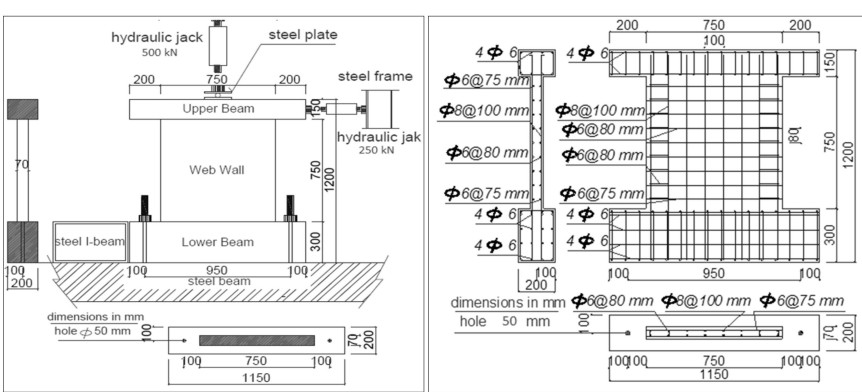

**Figure 4.** Concrete dimension, hydraulic jacks location, and reinforcement details.

**Table 6.** Specimens details.

| Group Name | Specimen Name | Dim. of Wall L × W × t (mm, mm, mm) | Opening Dim. and Loc. | | | Fiber Dimension mm |
| | | | a = b mm | X mm | Y mm | |
|---|---|---|---|---|---|---|
| N | Control | 750 × 750 × 70 | No Opening | | | No Fiber |
| B | SW-L/4-B | 750 × 750 × 70 | 187.5 | 375 | 93.75 | 1.5 b = 281.25 |
| | SW-L/3-B | 750 × 750 × 70 | 250 | 375 | 125 | 1.5 b = 375 |
| | SW-L/2-B | 750 × 750 × 70 | 375 | 375 | 187.5 | 1.5 b = 562.5 |
| M | SW-L/4-M | 750 × 750 × 70 | 187.5 | 375 | 375 | 2 b = 375 |
| | SW-L/3-M | 750 × 750 × 70 | 250 | 375 | 375 | 2 b = 500 |
| | SW-L/2-M | 750 × 750 × 70 | 375 | 375 | 375 | 2 b = 750 |
| T | SW-L/4-T | 750 × 750 × 70 | 187.5 | 375 | 281.25 | 1.5 b = 281.25 |
| | SW-L/3-T | 750 × 750 × 70 | 250 | 375 | 625 | 1.5 b = 375 |
| | SW-L/2-T | 750 × 750 × 70 | 375 | 375 | 562.5 | 1.5 b = 562.5 |

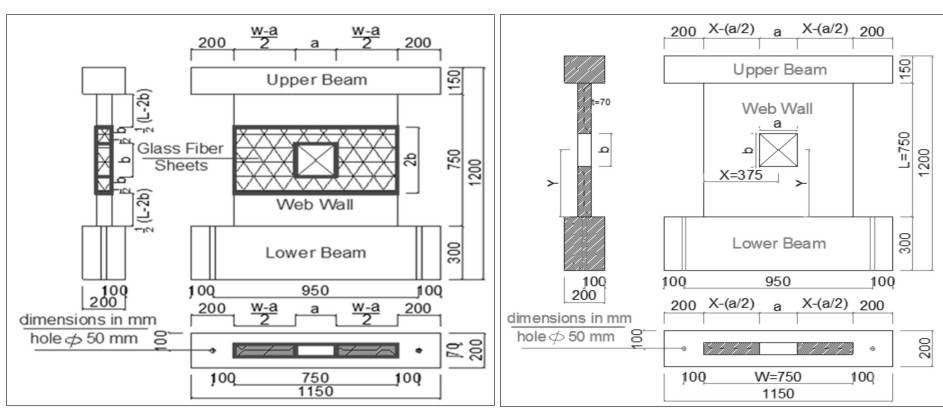

**Figure 5.** Geometry of the Wall, and the strengthening scheme for shear wall with middle opening.

### 3.3. Casting and Testing Procedure

Figure 6 describes the preparation of wooden frames and the steel reinforcement cages. The vertical and horizontal reinforcement was designed by the recommendation of the ECP-203 [2], and more horizontal stirrups were added to confine the wall edges and enhance strength and ductility. Electrical strain gauges with lengths of 10 mm and 120 Ohms resistance were attached to the reinforcing bars, as demonstrated in Figure 7. Three linear variable differential transformers (LVDTs) were utilized to measure displacement at various locations. The specimens' shape before and after strengthening is illustrated Figure 8. After the working load was applied, the cracks of the walls were repaired, and then, the FRP sheet was installed.

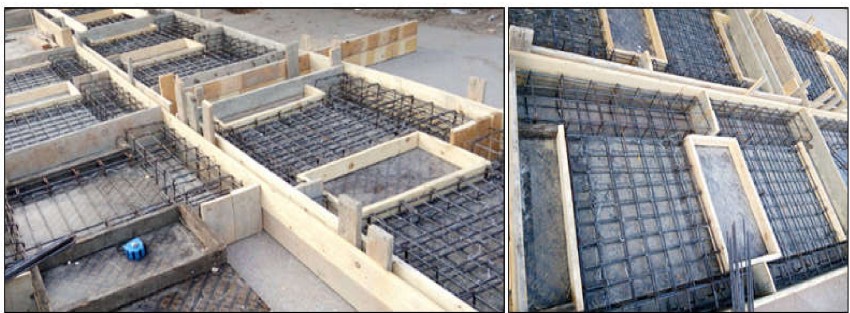

**Figure 6.** Preparation of the wooden forms and the reinforcement cages.

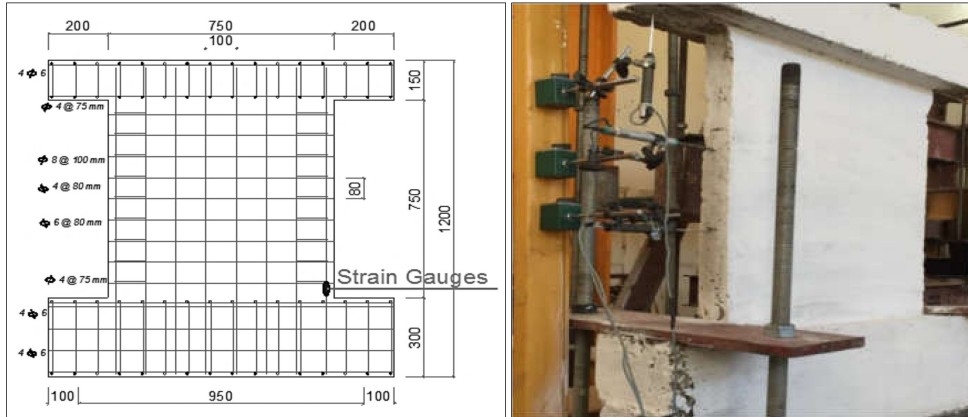

**Figure 7.** Strain gauge and dial gauge location.

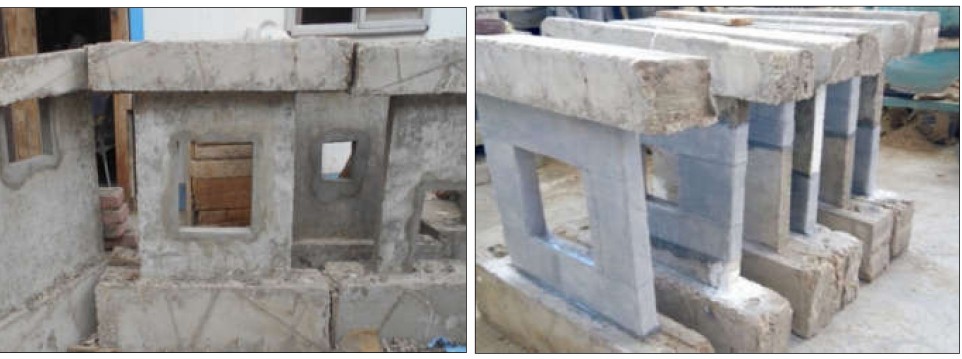

**Figure 8.** Casted shear wall before and after strengthening by GFRP sheets.

To determine the working load value for each specimen, the control-solid shear wall and the un-strengthened shear wall specimen with an opening (SW-L/4-M) were loaded to failure. Both results were then compared to determine the load capacity loss after installing the opening, and these results served as a baseline for choosing working load values of other un-strengthened walls with various openings. Furthermore, the findings of Behfarnia et al. [39] were helpful for estimating the working load values; for walls with an opening area of 6.25%, the working load was about (50–75%) of the ultimate load of control shear wall; for walls with an opening area of 11.11%, the working load was approximately (40–50%) of ultimate load of control shear wall; and for walls with an opening area of 25%, the working load was about (30–40%) of ultimate load of control shear wall, and considering the above-mentioned data and the appearance of developed diagonal cracks on each wall specimen, the working load value was carefully selected.

The openings were created, and the steel reinforcement was intercepted by the inserted openings, and then, the edges of openings were sharpened using cement mortar. Digital load cells of capacity 500 kN and 250 kN were adopted to measure the vertical load and the horizontal load, respectively. Steel frames are used to serve as a support for horizontal and vertical jacks. These steel frames are commonly used as lateral retaining structures, and they are expected to prevent the shear wall from deforming out of the plane.

The testing procedure started with a constant vertical load equal to 50 kN, and then, the lateral load was applied gradually by a 250 kN manual hydraulic jack that is mounted to the top beam of the shear wall, with a loading rate of about 15–25 kN per step until the failure occurred. By using a monitor connected to the load cell, the load values were recorded on the paper sheet at every increment of the load. Figure 9 demonstrates load cells, dial gauge, and test setup.

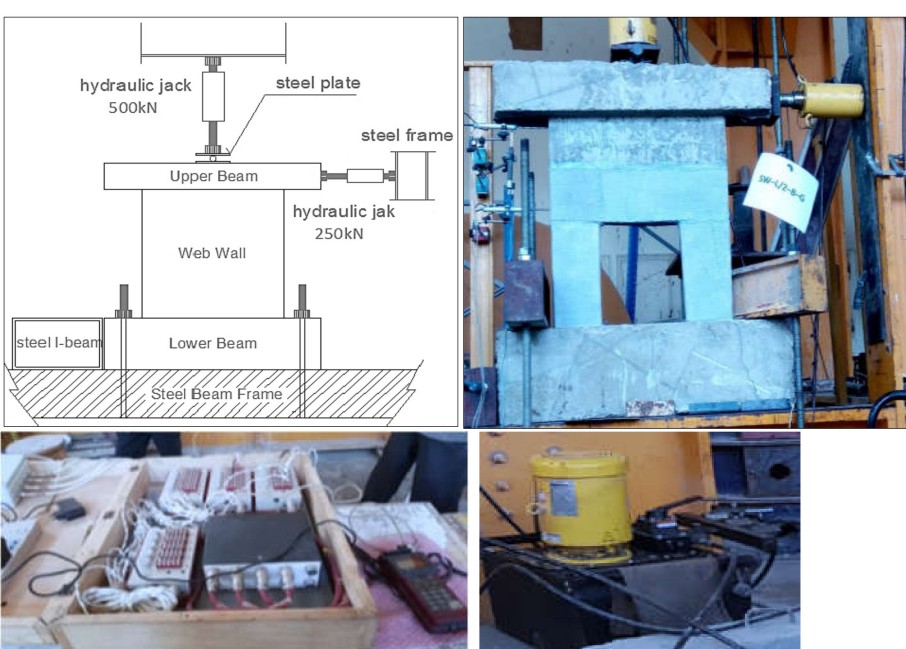

**Figure 9.** Load cells and test setup.

## 4. Experimental Results

### 4.1. Load Capacity

Table 7 presents the first cracking load, working load, and their corresponding deflections before strengthening and failure loads after strengthening. In fact, there are different levels of failure, and each of them can be a form of failure. When conducting experimental tests, the technique would not continue until the wall was entirely destroyed and collapsed. The test would often continue until a peak strength decrease of between 75–80% was noticed, or it would be stopped if the test specimen exhibited the concrete spalling mode of failure [38,39]. In our research, the failure loads occurred either by the failure of GFRP sheets or by concrete crushing.

**Table 7.** Loads and crack pattern of tested shear walls.

| Group | Specimen Name | Before Strengthening | | | | After Strengthening | | Failure Mode | |
|---|---|---|---|---|---|---|---|---|---|
| | | Cracking | | Working | | Failure | | | |
| | | $P_{cr}$ kN | $\Delta_{cr}$ mm | $P_w$ kN | $\Delta_w$ mm | $P_F$ kN | $\Delta_F$ mm | Before Strengthening | After Strengthening |
| N | Control | 64.28 | 0.76 | 190.0 | 7.63 | - | - | Shear Failure | —— |
| B | (SW–L/4–B) | 60.97 | 2.01 | 142.8 | 11.57 | 172.27 | 8.911 | Shear Failure | Fracture of GFRP |
| | (SW–L/3–B) | 40.55 | 1.06 | 90.99 | 4.49 | 148.71 | 7.823 | Shear Failure | Fracture of GFRP |
| | (SW–L/2–B) | 41.97 | 1.92 | 61.15 | 3.87 | 86.84 | 6.758 | Shear Failure | Fracture of GFRP |
| M | (SW–L/4–M) | 62.04 | 1.91 | 101.05 | 4.50 | - | - | Shear Failure | Fracture of GFRP |
| | (SW–L/3–M) | 35.22 | 1.26 | 80.00 | 8.96 | 147.88 | 8.596 | Shear Failure | Fracture of GFRP |
| | (SW–L/2–M) | 31.02 | 0.98 | 56.06 | 3.65 | 91.76 | 12.951 | Shear Failure | Fracture of GFRP |
| T | (SW–L/4–T) | 58.70 | 1.84 | 118.63 | 3.09 | 170.73 | 8.778 | Shear Failure | Fracture of GFRP |
| | (SW–L/3–T) | 26.69 | 1.54 | 106.32 | 7.67 | 160.60 | 9.837 | Shear Failure | Fracture of GFRP |
| | (SW–L/2–T) | 44.40 | 6.61 | 54.00 | 10.76 | 105.81 | 14.922 | Shear Failure | Fracture of GFRP |

Figure 10 shows the impact of the cut-off size and location on the lateral-load capacity of the tested structural walls. Generally, the larger the opening size, the lower the load capacity of the shear wall even with the strengthening of the shear wall around the openings; moreover, the shear walls with an opening at the mid-height have smaller load capacities despite the bottom and top opening. This was expected because the central panel at the middle height of the squat RC shear wall is considered critical section at which the strut-

and-tie action is formed to resist applied axial and lateral loads [5]; therefore, making an opening in this location causes interruption in the load paths, stress distribution, and force transfer within the wall.

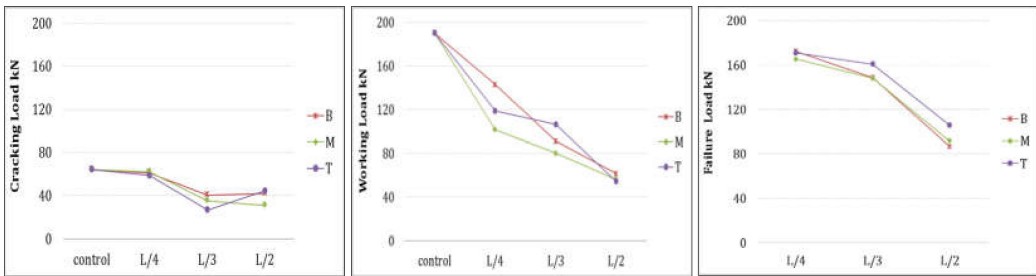

**Figure 10.** Effect of varying opening size and location on load capacity.

Turning to details, for three studied locations, creating square openings with an area equal to 6.25% of the total wall area resulted in a reduction in load capacity in the range of 24.84–46.82%, creating square openings with an area equal to 11.11% of the total wall area resulted in a load capacity reduction of about (44.04–57.89%), and creating square openings with the area equal to 25% of the total wall area resulted in a decrease in lateral load capacity of about (67.82–71.58%). After strengthening, the 6.25% opening caused a reduction in load capacity by (9.33–22.17%), the 11.11% opening caused a reduction in load capacity by (15.47–21.73%), and the 25% opening caused a reduction in load capacity by (44.31–54.29%).

It is also worth pointing out that, the load capacities of all retested strengthened RC shear walls increased by a percentage ranging between (42.01–95.94%), and their displacement capacities enhanced, as the displacement level of the strengthened RC walls was higher than those without strengthening.

Despite GFRP sheets' effectiveness in improving load and displacement capacity, this positive contribution was still insufficient to bring it up to the capacity of RC shear walls without openings. This is a result of post-construction cut-out openings, especially those with large sizes, which weaken the integrity and stiffness of the wall by reducing the resistant concrete and the reinforcement ratio as well as affecting the reinforcement configuration.

### 4.2. Crack Pattern and Failure Type

Figure 11 shows the crack development of the ten tested shear walls before strengthening. For the control solid shear wall, the first crack was a tension crack of the concrete that appeared horizontally at the right bottom of the wall due to an applied lateral load of 65 kN, and this cracking pattern is usually the same for all squat solid shear walls [35]. The effect of these cracks on wall stiffness is very small and can be ignored [40]. As the lateral load increased, the first diagonal crack was formed, and then, the tension-steel yielded and caused a larger diagonal crack (corner-to-corner), and this developed diagonal cracks resulting in diagonal shear failure at a load of 190 kN. The presence of the top beam helps in redistributing the shear load and controls the sudden failure after the development of the cracks, and this failure mechanism matches with the research of Paulay et al. [41], who studied the behavior of squat shear walls.

For un-strengthened RC walls with an opening, the first cracks appeared horizontally and diagonally due to the applied lateral load, and then, cracks began around the opening and propagate to the wall sides as the lateral load increased. Finally, all specimens tended to fail under the influence of shear.

Two main possible failure modes were reported for the bonding interface between the FRPs sheet and the concrete substrate: either material failure modes (which depend on the material strengths of the concrete and the FRP composite) or premature interface debonding failure modes (which depend on the bond between concrete and FRP sheet) [42].

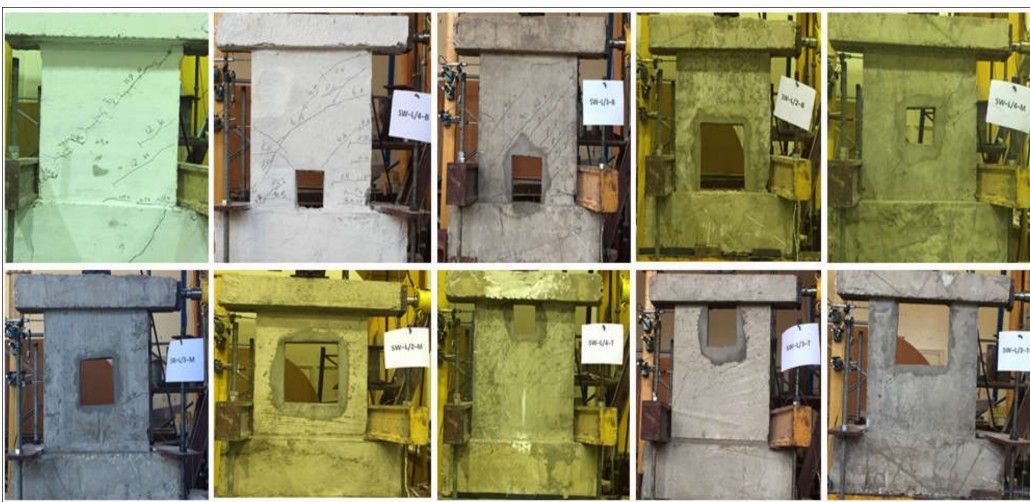

**Figure 11.** Crack pattern of shear wall before strengthening.

After strengthening specimens with GFRP sheets and reloading, the developed cracks and failure type in shear walls were dependent on the interaction between concrete, reinforcement, and the GFRP sheet as well as the sample's opening size and location. From Figure 12, it can be seen that most of the shear walls failed due to the material failure mode, and this failure type has three shapes for shear walls with small openings with areas 6.25% and 11%: the first shape was GFRP sheet rupture (this type typically occurs near the bottom corners of the opening due to stress concentration), the second was a crush of concrete within the compressive zone (it was obvious in the shear wall with a top opening; in this case, spalling was noticed in both sides of the specimen near the foundation), and the third was a shear failure. However, in shear walls with a 25% opening area, the wall behavior changed to a frame action, where overturning moments are resisted by an axial compression-tension coupled across the wall piers rather than by the individual flexural or shear action of the walls [38].

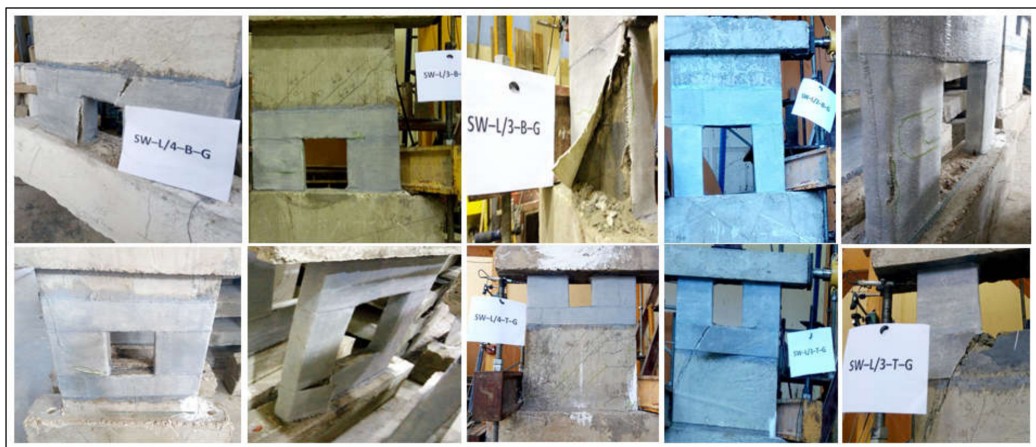

**Figure 12.** Failure modes of strengthened shear wall.

*4.3. Lateral Load versus Displacement Behavior*

(LVDT) were used to measure lateral displacement; Figures 13 and 14 illustrate the relationship between the applied lateral loads versus the horizontal displacement at the top of the described tested shear walls before and after strengthening, respectively. From the figures, it can be seen that the load-deflection curves of all specimens behave in the same manner. Initially, the load-deflection curve remained elastic till reaching the cracking load, and then, the relation started to be more curved until reaching the ultimate load and, finally, the failure stage. At the three studied locations, the slope of the curve decreased

as the opening size increased owing to the degradation in the stiffness and rigidity. After strengthening, the slope of the curves and the lateral load capacity increased due to the ability of the applied GFRP sheets in controlling stress redistribution and crack propagation. The displacement capacity of the walls with the opening was lesser than that of the reference wall without opening, but it was improved by applying GFRPs around the opening; in this way, a much more ductile load-displacement response can be obtained.

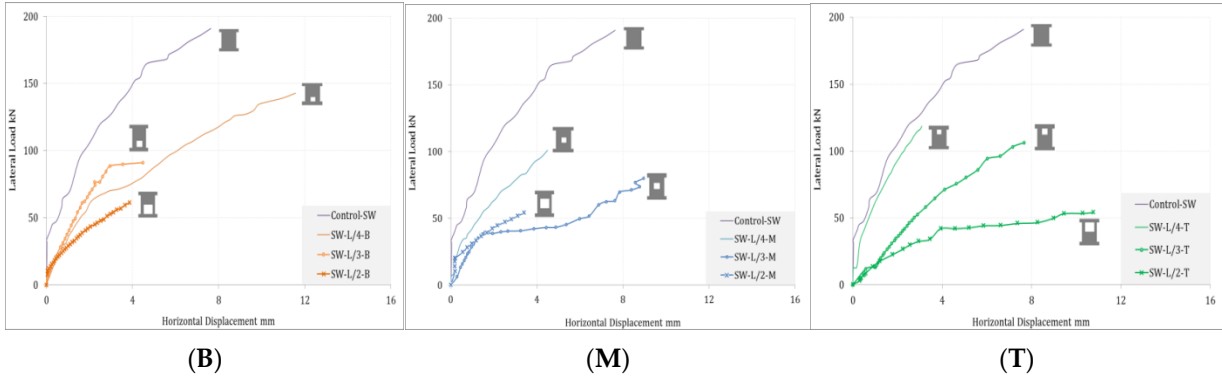

**Figure 13.** Lateral load versus lateral displacement curves for groups (**B**), (**M**), and (**T**) before strengthening.

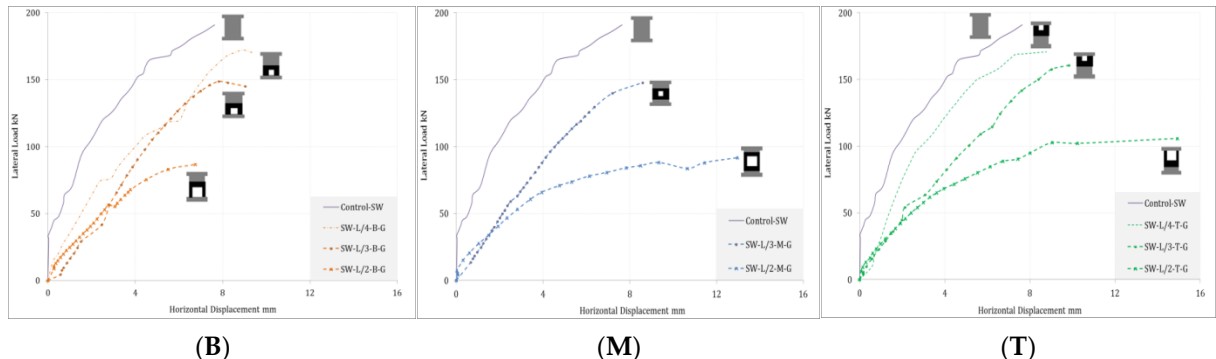

**Figure 14.** Lateral load versus lateral displacement curves for groups (**B**), (**M**), and (**T**) after strengthening.

*4.4. Reinforcement Strain*

As described before, the strain gauge was fixed on the vertical bar at the maximum tensile direction to measure the axial strain in the reinforcement bars. The data were recorded using a data-acquisition system. Figures 15 and 16 demonstrate the relationship between lateral load and the monitored strain before and after retrofitting. The yield strain and ultimate strain for used tension-steel bars could be calculated as follows: $\varepsilon_y = F_y/E_y = 280/200{,}000 = 0.0014$ and $\varepsilon_u = F_u/E_u = 380/200{,}000 = 0.0019$. Before cracking, the shear resistance was carried by concrete. Once the cracks developed, the vertical tension reinforcement started to record strains for the control shear wall specimen; the maximum strain achieved by the tension bar was about 0.0021, which means this wall reached its maximum strain before strengthening, and the ultimate strain was attained in the tension bar of the walls with an opening area equal to 6.25 percent of the total wall, particularly for the walls with openings at the bottom and middle. As the opening size increased, the ultimate strain value decreased, and the main tension reinforcements reached their yield point. From Figure 16, it is clear that the strengthening of walls does not have a significant effect on the strain of tension reinforcement despite increasing the load capacity, and these results are similar to that described by Volnyy et al. [17]: the value of the observed strain at failure for the walls with 11.11% opening size were about (0.0009–0.0012) and for the walls with 25% opening size were about (0.001–0.0015). This occurs due to the post-constructed cut-out openings in the shear walls, especially those with large sizes, which affect the reinforcement arrangements, decrease the reinforcement ratios, and form

vertical elements (RC short column) next to the opening. Therefore, these elements may not have enough confinements or shear reinforcement to significantly affect the strain of tension reinforcement. Additionally, given the slenderness of the shear wall, since the height/length ratio is 1, the specimens are not flexure-critical by shear-critical.

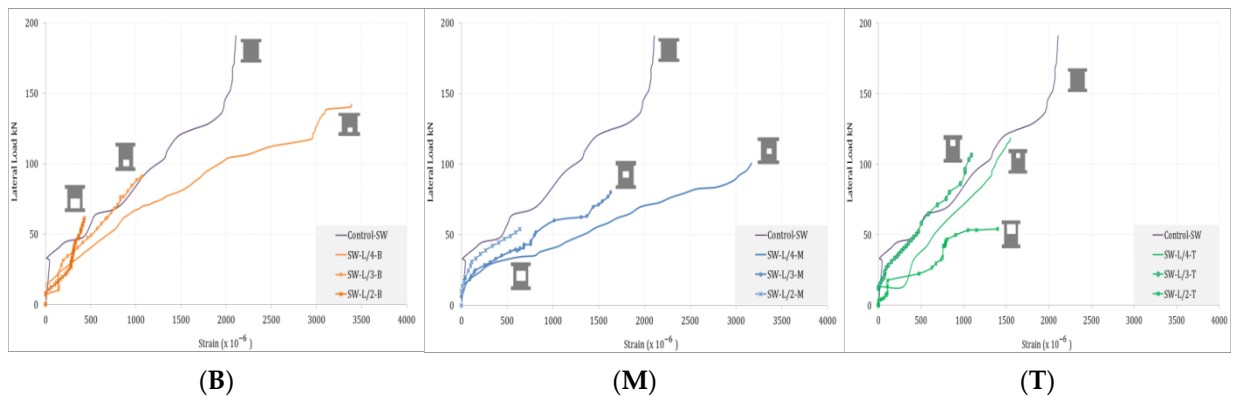

**Figure 15.** Lateral load-strain relationships for all tested walls before strengthening.

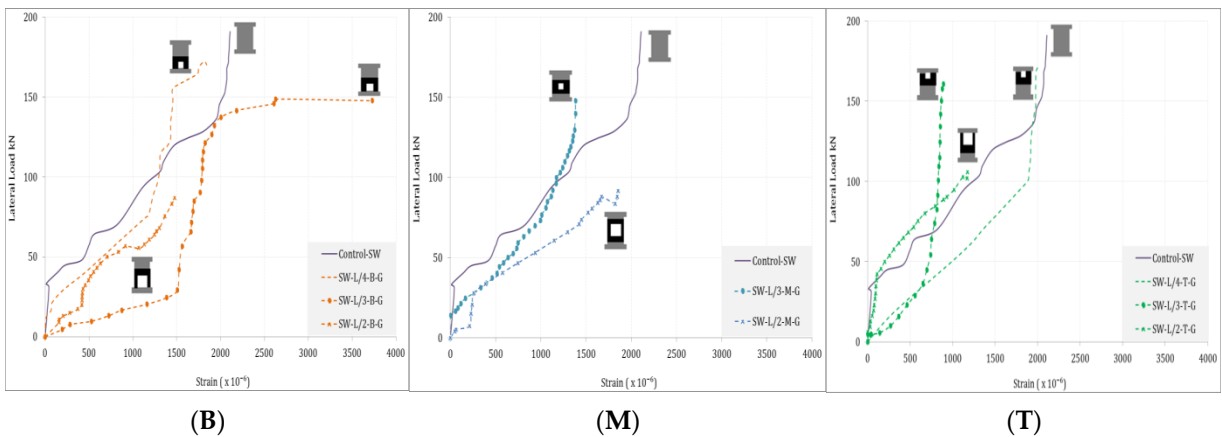

**Figure 16.** Lateral load-strain relationships for all tested walls after strengthening.

*4.5. Lateral Stiffness*

Stiffness is the load required to cause a unit deflection; un-cracked lateral stiffness $K_i$ (initial stiffness) of the tested walls before and after strengthening were calculated and are reported in Table 8. The un-cracked stiffness ($K_i$) is defined as the slope of the lateral load-displacement curve at a load value less than the cracking load. The initial stiffness of the reference wall was 84.58 kN/mm, and as the opening size increases, the initial stiffness decreases. Whereas the initial stiffness of strengthened walls was smaller than un-strengthened walls, and this was expected due to using the same tested walls that had cracks, these cracks caused a drop in the initial lateral stiffness, but the strengthened specimens had delayed stiffness degradation compared with un-strengthened ones.

As shown in Figure 17, when comparing shear walls with the smallest opening area of 6.25%, the shear wall with the top opening has more stiffness than the walls with a middle and bottom opening; this is because the top opening is quite far from the load-transfer paths, and by placing the opening at the top of the wall, the struts were more ideally formed, and the transmitting flows of the internal compression load were more similar to the control specimen.

**Table 8.** Lateral stiffness, ductility, and absorbed energy of the tested walls.

| Specimen Name | Initial Stiffness Ki (kN/mm) | | Ductility (mm) | | Absorbed Energy (kN·mm) | |
|---|---|---|---|---|---|---|
| | Before Strengthening | After Strengthening | Before Strengthening | After Strengthening | Before Strengthening | After Strengthening |
| Control | 84.58 | —— | 3.356 | —— | 1023 | —— |
| (SW–L/4–B) | 30.33 | 33.42 | 6.232 | 6.013 | 1068 | 985.5 |
| (SW–L/3–B) | 36.73 | 21.89 | 2.011 | 4.779 | 289.2 | 805.2 |
| (SW–L/2–B) | 28.40 | 20.79 | 1.921 | 3.337 | 152.7 | 383.8 |
| (SW–L/4–M) | 32.48 | —— | 2.322 | —— | 287.2 | —— |
| (SW–L/3–M) | 28.72 | 22.88 | 6.865 | 4.695 | 412.4 | 755.4 |
| (SW–L/2–M) | 25.96 | 21.04 | 1.454 | 3.930 | 127.3 | 900.6 |
| (SW–L/4–T) | 48.21 | 35.97 | 1.702 | 3.895 | 231.1 | 1008 |
| (SW–L/3–T) | 17.37 | 23.11 | 4.630 | 6.220 | 467.8 | 918.9 |
| (SW–L/2–T) | 11.02 | 21.20 | 3.447 | 4.937 | 408.7 | 1209 |

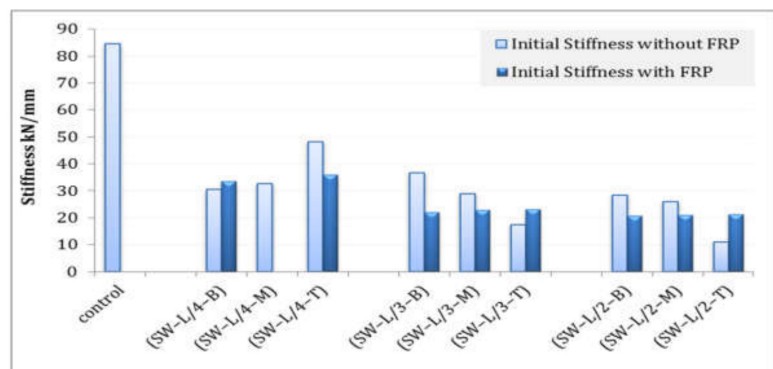

**Figure 17.** Initial stiffness of all tested shear walls with and without GFRP sheets.

On the other hand, as the opening size increases, stress flow disturbance increases, and stiffness decreases. Additionally, the stiffness value was highest when the opening was at the base of the wall because the shear walls exhibited frame action behavior.

*4.6. Ductility and Absorbed Energy*

Ductility is defined as the ability of RC members to sustain considerable deformation before failure with maintaining suitable load capacity [43]. This characteristic is considered essential especially in seismic areas, as it provides signs of failure and prevents brittle failure. Furthermore, ductility measures the ability of a concrete member to dissipate energy when tested to failure [44]. Two methods to evaluate ductility are usually used. The first one is the ratio of the ultimate deflection to the deflection at yield. The second one is the deflection at 70% or 80% of the maximum loads to ensure the reduction in stiffness due to cracking near the end of the elastic stage [36]. The brittle behavior of squat shear walls and creating openings caused the main tension reinforcement to not yield in most of the tested specimens. Therefore, the second method was used in this research since the deflection at about 75% of the corresponding working load for un-strengthened walls and the deflection at about 75% of the corresponding failure load for strengthened walls were used to calculate the ductility. The results are listed in Table 8.

The results reveal that increasing the opening size leads to a decrease in the ductility in all three studied positions. After strengthening, the lateral displacement of most tested walls increased, and accordingly, the ductility increased. The increased ductility of strengthened sections resulted from the confining stresses that enable concrete to develop more compressive strains before the failure [45].

The energy absorption is calculated as the area under the lateral load-deflection curve till ultimate failure load and given in Table 8, so the area under lateral working-load-deflection for un-strengthened shear walls and the area under lateral failure-load-deflection for strengthened shear walls were estimated.

The findings show that the strengthened RC walls continued to sustain additional inelastic deformations before failure due to the confinement effect provided by the GFRP sheets that enhanced the wall ductility and absorbed energy. A comparison between ductility and absorbed energy values of all tested shear walls is shown in Figure 18.

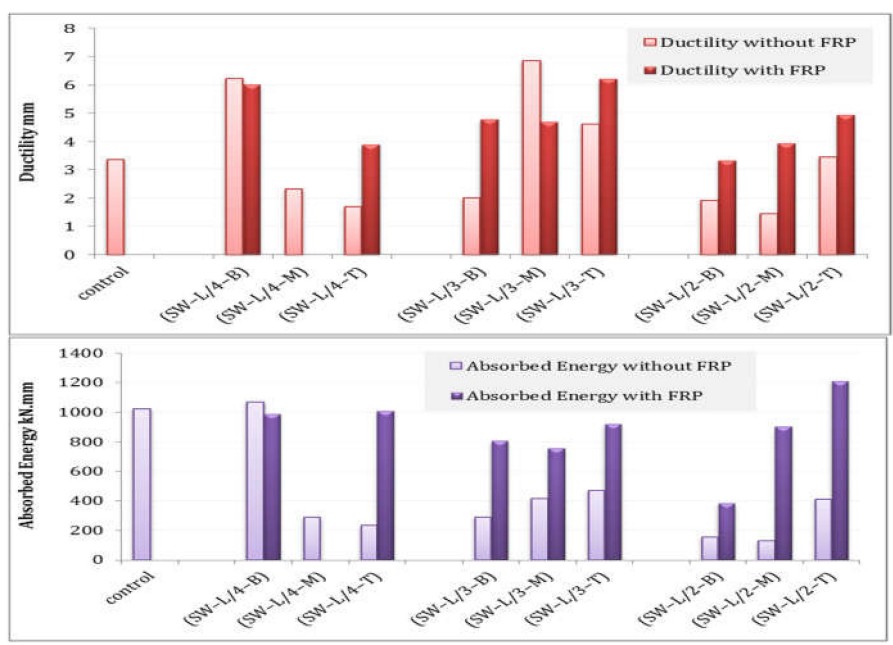

**Figure 18.** Ductility and absorbed energy of all tested shear walls with and without GFRP sheets.

## 5. Finite-Element Modeling (FEM)

Finite-element analysis (FEA) has been widely used in the last decades to predict the behavior of the structural element and investigate critical aspects such as cracking patterns, crushing of concrete, induced stresses, etc. Using FEA promotes sustainability since it enables the testing of structural elements before the construction stage, thereby reducing material consumption and saving the time and cost required for design and analysis with reliable accuracy.

In this research, the FE models were established by ANSYS 14.5 package; the nonlinear FE models were developed using proper elements for concrete, steel reinforcement, loading plates, GFRP sheet, and interface element.

### 5.1. Pre-Processing

#### 5.1.1. Element Type

The eight-node solid-element SOLID65 was used to model the concrete; LINK180 was used to discretely model the steel reinforcement bars; the GFRP sheet was modeled using SHELL181; for steel plates, SOLID185 element was used; and COMBIN39 element used to simulate bond-slip between concrete and GFRP sheet interface. All of these elements were used successfully by [46–50].

#### 5.1.2. Real Constant

For the LINK180 element, two cross-section areas were defined ($0.503$ mm$^2$ and $0.283$ mm$^2$) to represent horizontal, vertical, and stirrups reinforcement bars used in the RC walls. For the COMBIN39 element, the load-displacement curve was added using equations of Lu et al. [51], as shown Table 9.

**Table 9.** Real constant for COMBINE39 element.

| Real Constant Set No. 4, COMBIN39 Element | | |
|---|---|---|
| Displacement-Force Curve Data | Displacement (mm) | Force (MPa) |
| Data set 1 | 0.00 | 0.00 |
| Data set 2 | 0.0056 | 25.00 |
| Data set 3 | 0.056 | 0.00 |

### 5.1.3. Material Properties

The nonlinearity in static FEA occurs either due to material nonlinearity or geometric nonlinearity or both (ANSYS, Inc.) [52], in this research, the material nonlinearity was considered due to cracking and crushing of concrete and steel yielding.

SOLID65 element for concrete requires linear isotropic and multi-linear isotropic material properties, as shown in Table 10. $E_X$ is the modulus of elasticity of the concrete, and PRXY is the Poisson's ratio ($\nu$), and it was assumed to be 0.2 for all specimens. The elastic modulus was calculated based on the Egyptian code equation (ECP 203, 2020) [2] $E_c = 4400\sqrt{f_{cu}}$ MPa. As shown in Figure 19, the compressive uniaxial stress–strain relationship for the concrete model was obtained using the Desayi et al. [53] equation:

$$f = E_c * \varepsilon / \left[ 1 + \left( \frac{\varepsilon}{\varepsilon_o} \right)^2 \right] \tag{1}$$

$$\varepsilon_o = 2f_c' / E_c \tag{2}$$

$$E_c = f / \varepsilon \tag{3}$$

**Table 10.** Material properties for concrete (SOLID65 element).

| Linear Isotropic | | Multi-Linear Isotropic | | Concrete | |
|---|---|---|---|---|---|
| EX | 24,580 | Stress | Strain | $\beta o$ | 0.30 |
| PRXY | 0.20 | 0.0006 | 13.51 | $\beta c$ | 0.90 |
| | | 0.009 | 18.41 | $fr$ | 3.34 |
| | | 0.0012 | 21.76 | $fc'$ | 25.00 |
| | | 0.0015 | 23.72 | | |
| | | 0.002 | 24.80 | | |
| | | 0.003 | 25.00 | | |

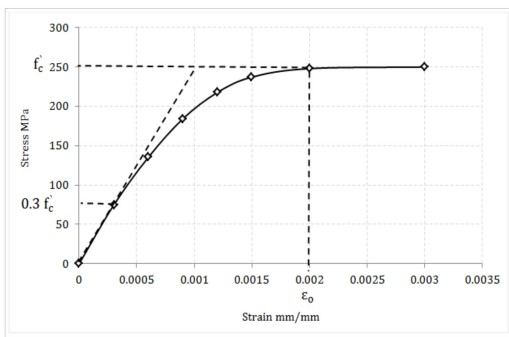

**Figure 19.** Stress–strain curve for concrete.

Concrete material model also requires different constants to be defined, and these constants include the uniaxial tensile cracking stress ($f_{ct} = 3.34$ MPa), the uniaxial crushing stress ($f_{cu} = 31.2$ MPa), and shear transfer coefficients for open crack ($\beta_o = 0.3$) and

for closed crack ($\beta_c = 0.9$). The values of these coefficients were chosen as suggested by Kachlakev et al. [54].

LINK180 element for steel reinforcement requires linear isotropic and bi-linear isotropic material properties, as shown in Table 11 with the following data: elastic modulus ($E_s = 200$ GPa), Poisson's ratio ($\upsilon = 0.30$), and yield stress ($f_y = 280$ MPa). The elastic-perfectly plastic model was used, as shown in Figure 20, with a perfect bond between reinforcement and concrete.

**Table 11.** Material properties for steel bars and loading plates.

| | Linear Isotropic | | Bilinear Isotropic | |
|---|---|---|---|---|
| **4. LINK180 (Steel Bars)** | EX | $2 \times 10^5$ | Yield Stress | 280 |
| | PRXY | 0.30 | Tang Mod | 0.00 |
| **5. SOLID 185 (Loading Plates)** | Linear Isotropic | | | |
| | EX | $2 \times 10^5$ | | |
| | PRXY | 0.30 | | |

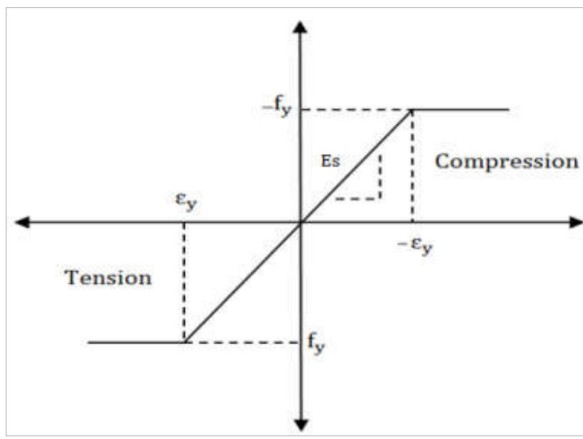

**Figure 20.** Bilinear stress–strain curve of steel reproduced from [51].

SOLID185 element is used to model loading plates; this element is modeled as a linear isotropic with the steel modulus of elasticity ($E_s = 200$ GPa), and Poisson's ratio ($\upsilon = 0.30$).

SHELL181 element is used to model the GFRPs sheets; the data required for modeling this element involve the shell thickness and orthotropic material properties, as listed in Table 12.

**Table 12.** Material properties of the GFRP sheets.

| FRP Composite | Elastic Modulus MPa | Poisson's Ratio | Shear Modulus MPa | Tensile Strength MPa | Thickness of Sheet mm |
|---|---|---|---|---|---|
| | $E_x = 72{,}000$ | $\upsilon_{xy} = 0.30$ | $G_{xy} = 28{,}570$ | | |
| GFRP | $E_y = 72{,}000$ | $\upsilon_{xz} = 0.26$ | $G_{xz} = 1748$ | 2500 | 0.20 |
| | $E_z = 4545$ | $\upsilon_{yz} = 0.26$ | $G_{yz} = 1748$ | | |

Two methods were used to model the bond between concrete element SOLID65 and FRP element SHELL181; the first method assumed a perfect bond, and the coincident nodes between the concrete and FRP interfaces were merged with each other. Examples include the research conducted by [55–57], while the second method considered bond-slip, and the FRP/concrete interfaces were modeled using spring element with proper bond-slip law as shown in Figure 21. In this paper the second method was used to develop accurate nonlinear FEA of FRP-strengthened RC walls. COMBIN39 element is used as the interface element with a bi-linear bond-slip law. The bi-linear bond-slip model suggested by Lu et al. [51] is shown in Figure 22.

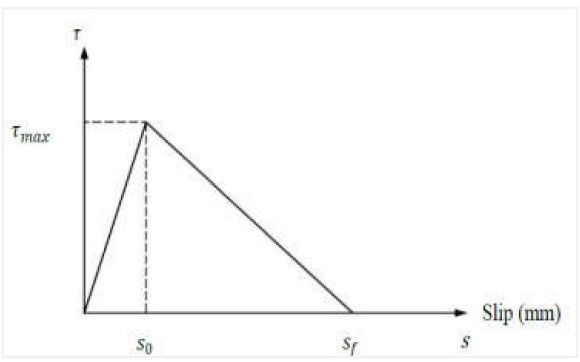

**Figure 21.** The bilinear bond-slip model reproduced from [51].

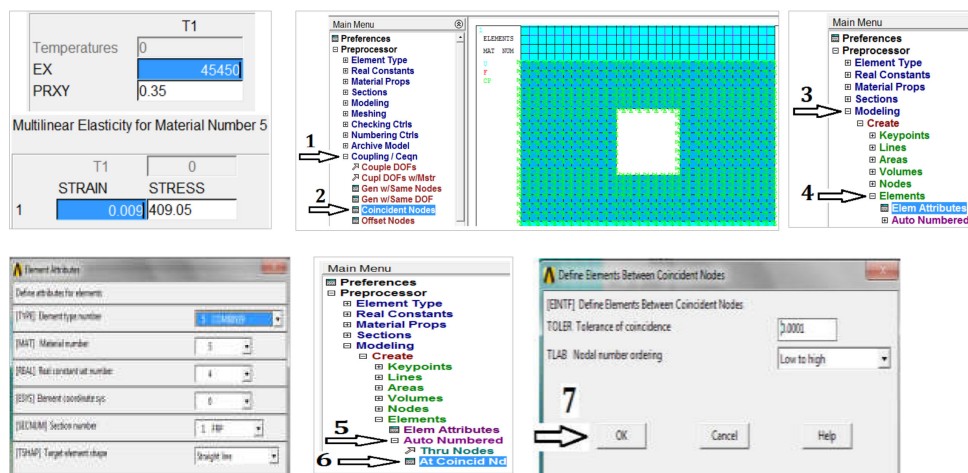

**Figure 22.** Material properties and modeling of COMBINE39 element.

The model has two stages, i.e., an elastic stage and a softening stage, and requires three main parameters: maximum bond strength $\tau_{max}$; slip corresponding to maximum bond strength, $S_0$; and slip corresponding to ultimate bond strength, $S_f$. The area under the bond-slip graph represents the interfacial fracture energy, $G_f$. The next equations show the values of these parameters:

$$\tau_{max} = \alpha_1\,\beta_w\,f_t \qquad (\alpha_1 = 1.5) \tag{4}$$

$$\beta_w = \sqrt{\frac{2.25 - b_f/b_c}{1.25 + b_f/b_c}} \tag{5}$$

$$G_f = 0.308\,\beta_w^2\,\sqrt{f_t} \tag{6}$$

$$S_0 = 0.0195\,\tau_{max}\beta_w \tag{7}$$

$$S_f = 2\,\tau_{max}\beta_w \tag{8}$$

where

$b_c$ : The width of the concrete (mm);
$b_f$ : The width of the FRP sheet (mm);
$f_t$ : Maximum tensile strength of concrete $(N/mm^2)$.

### 5.1.4. Creating Volume and Meshing

The beam, wall, and loading plates were modeled as volumes with the same aforementioned experimental dimensions; the discrete model was used to represent steel reinforcement [54], and thereby, the concrete and the reinforcement mesh share the same nodes. The GFRP sheet was modeled as area, and its thickness was defined using thin area section. The

accuracy of FEA depends on the mesh geometry and size, and studies by [58–60] indicated that the smallest element dimension is chosen based on the maximum aggregate size, so square element mesh with length 25 mm was used in this study. The overall volume and meshing of studied shear walls, reinforcement, and GFRP sheets are shown in Figure 23.

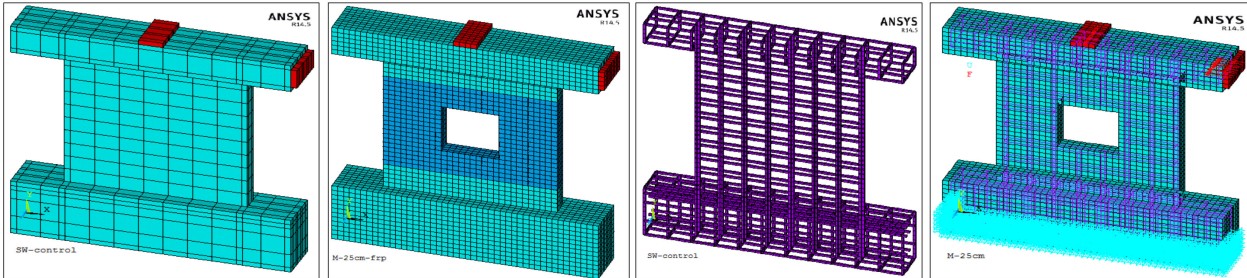

**Figure 23.** Volume, meshing, and boundary conditions of studied shear walls.

*5.2. Analysis Solver*

5.2.1. Applying Loads and Constrain

As shown in Figure 24, the axial and lateral forces (P) were applied throughout the entire center line of nodes of steel plates. The bottom side of the lower beam was constrained in X, Y, and Z directions with constant values of (0.0).

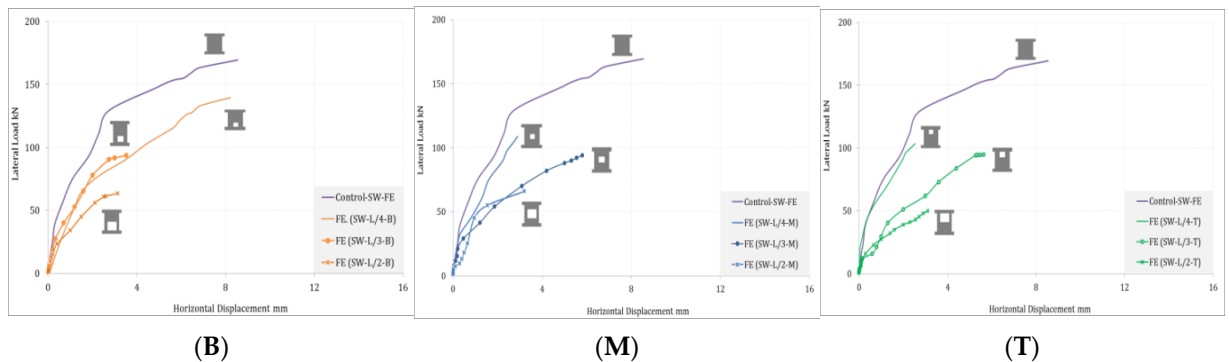

**Figure 24.** Lateral load-displacement curves from FEM and experimental work before strengthening.

5.2.2. Nonlinear Solution

In this paper, static analysis is used to determine reactions, crack patterns, stresses, etc. In nonlinear static analysis, stiffness is not constant. It is a function of displacement or material modulus or both. Through nonlinear static analysis, the load is applied in load increments. At every substep, ANSYS determines the difference between the applied loads and the loads corresponding to the elements' stresses; the stiffness matrix is adjusted at every iteration, and this method is known as Newton–Raphson method. The failure criteria of each model were identified in three possible ways: concrete crushing failure, debonding failure, and GFRP failure mode. When the solution for the load step does not converge, the program displays a message specifying that the models have a large deflection over the displacement limit of the ANSYS program [52]. The analysis of strengthened shear walls are conducted in two stages: first, without the use of GFRP sheets, the specimens are loaded from zero to working load, and the deflection is obtained at the end of this stage. The second stage is after applying GFRP sheets and includes the previously measured deflection, and then, the restart command is used to restart the analysis after the initial run has been completed [9].

### 5.3. Post-Processing

5.3.1. Lateral Load-Displacement Curves

In this stage, results such as lateral load-displacement curves, crack pattern, and stress contours could be obtained. Table 13 demonstrates a comparison between experimental results and FE results to validate the accuracy of the simulated FE models. In addition, the lateral load-displacement curves from experimental work were compared to those from the FE models, as shown in Figures 24 and 25, before and after strengthening, respectively. From the table and graphs, it was found that the maximum diffrence between failure experimental loads and FE ultimate loads were about (2–15%).

**Table 13.** Experimental results versus FE results.

| Specimen Name | Experimental Work | | | | FEM by ANSYS | | | | Before Strengthening | After Strengthening |
| | Before Strengthening | | After Strengthening | | Before Strengthening | | After Strengthening | | | |
| | $P_W$ kN | $\Delta_W$ mm | $P_F$ kN | $\Delta_F$ mm | $P_W$ kN | $\Delta_W$ mm | $P_F$ kN | $\Delta_F$ mm | $P_{W(Exp.)}/P_{W(FE)}$ | $P_{F(Exp.)}P_{F(FE)}$ |
|---|---|---|---|---|---|---|---|---|---|---|
| Control | 190.0 | 7.63 | —— | —— | 169.5 | 8.56 | —— | | 1.12 | —— |
| (SW–L/4–B) | 142.8 | 11.57 | 172.27 | 8.91 | 139.7 | 8.21 | 176.96 | 9.12 | 1.02 | 0.97 |
| (SW–L/3–B) | 90.99 | 4.49 | 148.71 | 7.82 | 93.74 | 3.52 | 136.98 | 5.72 | 0.97 | 1.09 |
| (SW–L/2–B) | 61.15 | 3.87 | 86.84 | 6.76 | 63.50 | 3.13 | 81.20 | 5.52 | 0.96 | 1.07 |
| (SW–L/4–M) | 101.05 | 4.50 | —— | —— | 108.9 | 3.00 | 149.41 | 4.32 | 0.93 | —— |
| (SW–L/3–M) | 80.00 | 8.96 | 147.88 | 8.59 | 94.13 | 5.81 | 140.20 | 6.62 | 0.85 | 1.06 |
| (SW–L/2–M) | 56.06 | 3.65 | 91.76 | 12.95 | 65.87 | 3.20 | 100.00 | 5.31 | 0.85 | 0.92 |
| (SW–L/4–T) | 118.63 | 3.09 | 170.73 | 8.78 | 103.5 | 2.54 | 158.00 | 4.58 | 1.15 | 1.08 |
| (SW–L/3–T) | 106.32 | 7.67 | 160.60 | 9.84 | 94.80 | 5.64 | 147.00 | 6.96 | 1.12 | 1.09 |
| (SW–L/2–T) | 54.00 | 10.76 | 105.81 | 14.92 | 49.92 | 3.11 | 93.41 | 5.14 | 1.08 | 1.13 |

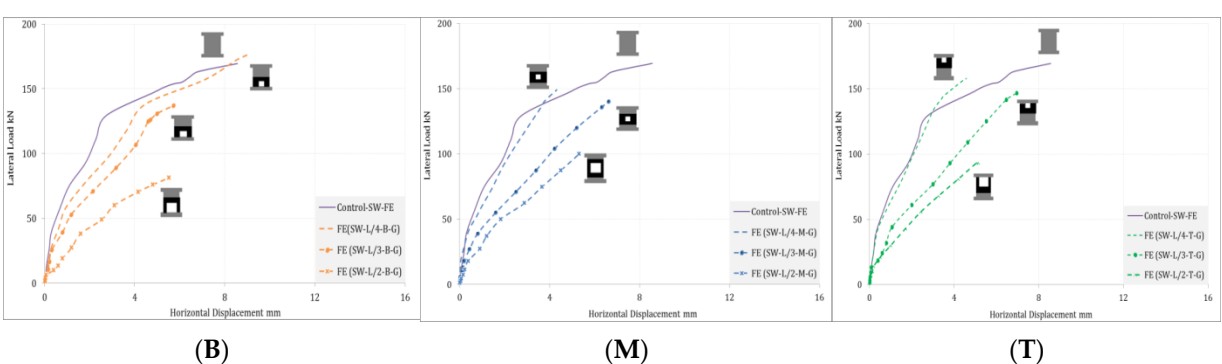

**(B)**　　　　　**(M)**　　　　　**(T)**

**Figure 25.** Lateral load-displacement curves from FEM and experimental work after strengthening.

It is worth mentioning that the experimental specimens have higher displacement values than finite-element models, and the two main reasons for that are as follows: the first is due to the post-constructed cut-out openings in the shear walls, particularly with large sizes, which affect the reinforcement arrangements, reduce the reinforcement ratios, and weaken the wall and, accordingly, decrease the stiffness and integrity of the wall as a whole; the second is that in the real test, the same shear walls were used before and after strengthening, but the FE models simulate shear walls using an approximate approach to account for the effect of reloading, which may result in higher stiffness and lower displacement at the same load.

Moreover, the lateral load-displacement curves from FEM are slightly stiffer than those from experimental work, and the higher stiffness of FE models is due to many factors, such as the micro-cracks that developed in actual walls from drying shrinkage, handling, and the crack branching process, which are not included in FEM [54]; additionally, using an idealized stress–strain curve for concrete and steel and assuming a perfect bond between concrete and steel resulted in higher stiffness for FE models.

Since the FE models are more homogeneous than the real walls, which have a variety of micro-cracks, a more detailed characterization of the concrete and steel materials' properties might lead to improved results for the predictions of the finite-element model. Testing core samples from the walls could be used to characterize the concrete. Instead of using design properties and an elastic-plastic model to characterize the steel, the tension of the steel bars could be tested to determine the actual stress–strain behavior and yield strength. Considering the findings and observations mentioned above, the FE models by ANSYS could be able to simulate the nonlinear behavior of RC shear walls, and they can also be used to investigate additional parameters to determine the shear capacity and provide design guidelines for strengthened squat SW with opening.

### 5.3.2. Deflection Contour, Cracks Pattern, and Stress Distribution

Figures 26–32 illustrate the deflection contours, cracks patterns, and stresses distribution at the last converged load step (at failure), and it was observed that:

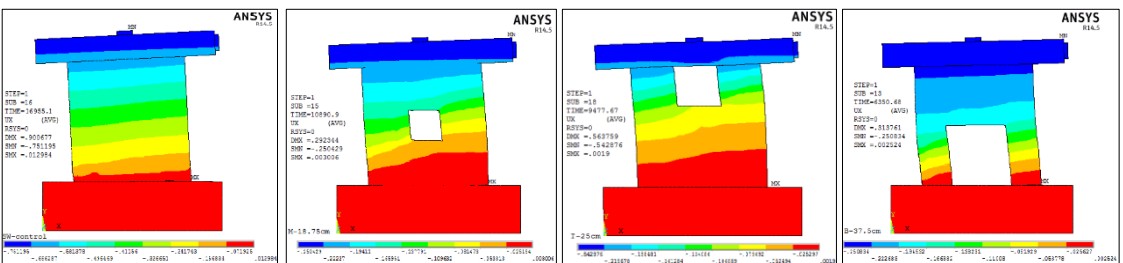

**Figure 26.** Deflection contours of some models.

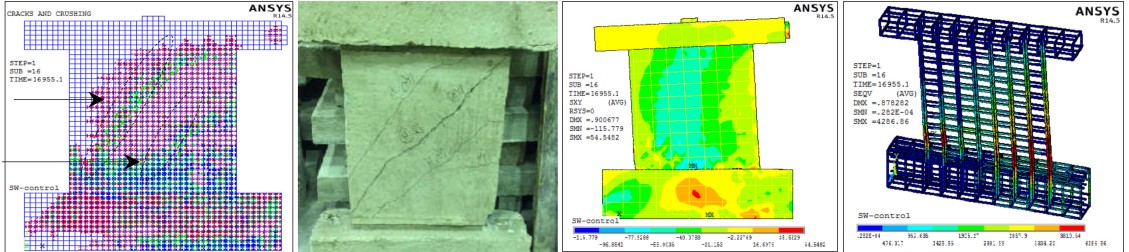

**Figure 27.** Crack pattern and stress contour of control-SW.

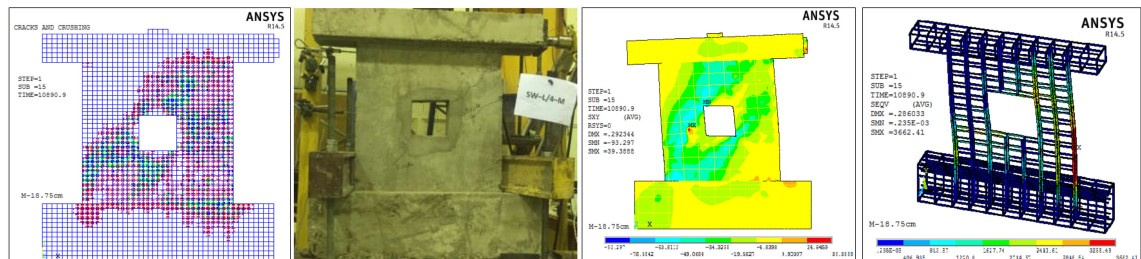

**Figure 28.** Crack pattern and stress contour of (SW-L/4-M) before strengthening.

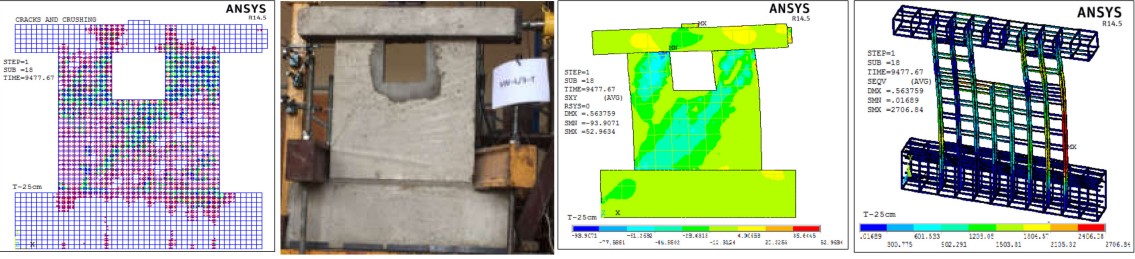

**Figure 29.** Crack pattern and stress contour of (SW-L/3-T) before strengthening.

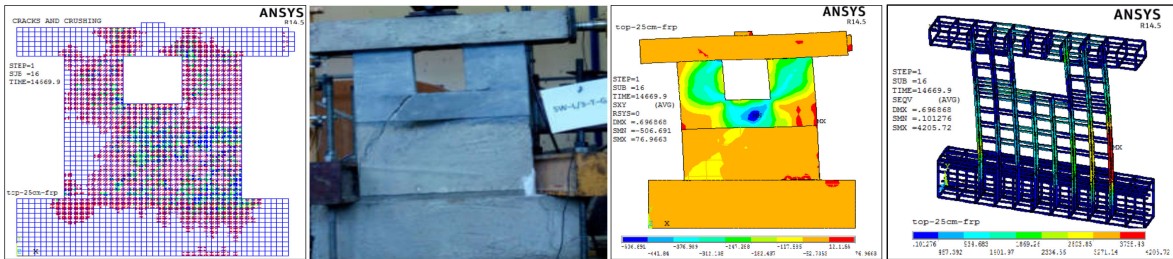

**Figure 30.** Crack pattern and stress contour of (SW-L/3-T) after strengthening.

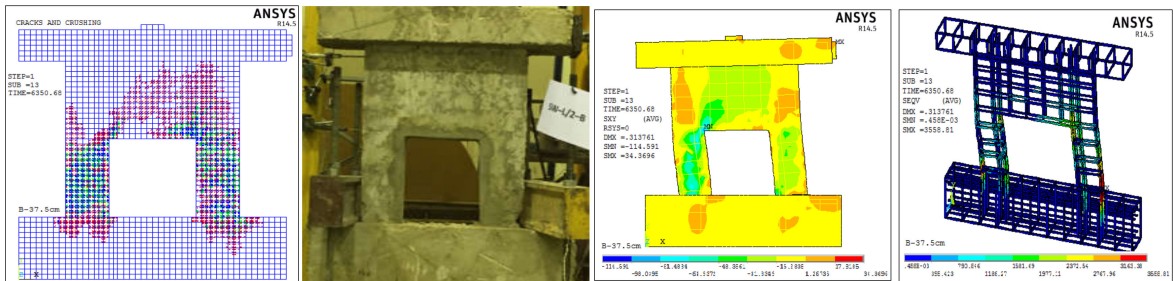

**Figure 31.** Crack pattern and stress contour of (SW-L/2-B) before strengthening.

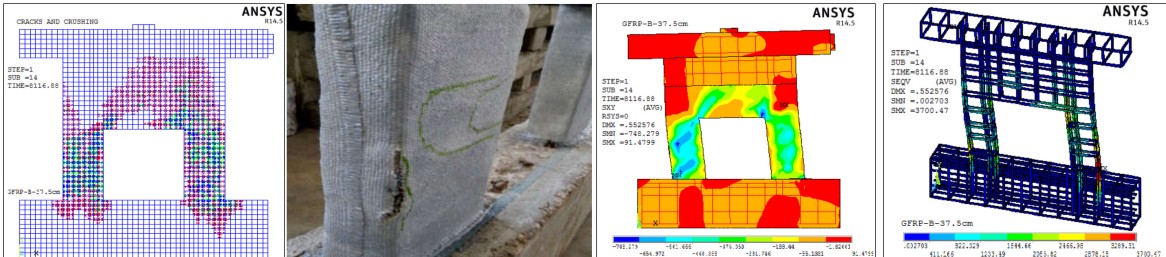

**Figure 32.** Crack pattern and stress contour of (SW-L/2-B) after strengthening.

All FE shear wall models have roughly the same deflection contours but with varying displacement values.

In ANSYS, the crack mode is recorded at each applied load step. When the principal tensile stress of concrete surpasses its ultimate tensile strength, the cracks appeared as circles perpendicular to the principal stress vector, and the crushing appeared as an octahedron outline [52]. The crack/crushing plot option was utilized with the vector mode plot turned on to display the cracks in SW models. From shown figures, the crack patterns of the SW models from FE fit the experimental patterns well and reflected their failure mode. For example, the crack pattern of FE control SW created diagonal cracks, which caused the diagonal shear failure. Additionally, the FE models with an opening exhibited the same crack pattern as the experimental ones, with cracks developing at the corner of opening then spreading away, which are consistent with the results of the experimental investigation. After strengthening, the FE wall with an opening showed a greater area with developed cracks, as the GFRP sheet aids in redistribution of stresses and allows more concrete parts to participate in carrying and transmitting the load.

The continuity of stress contour demonstrates that the mesh size is acceptable. Further, the stress distribution in concrete and steel are changed after applying the GFRP sheet, especially at the corner of the opening, as the steel bars gained more tension and compression stresses at the right and left external edges of the walls, respectively. Additionally, the highest shear stresses were developed around the opening and decreased outward, and these stresses' concentration was due to uneven straining of the adhesive and the eccentricity of the applied load, which resulted in the failure [61].

In Figure 28, for shear walls with small openings of areas (6.25% and 11%) and with mid-height position, the shear stress distribution reveals that two compressive struts were developed at the corner of openings parallel to the diagonal of the wall. Since mid-height openings disrupt the load paths, stress distribution, and force transfer within the wall, in contrast to the top and bottom openings, the compressive strut developed along the wall's diagonal, as shown in Figure 29.

For the walls with bigger opening sizes, the walls behave like frame structures, as shown in Figure 31. In addition, the maximum stress of the GFRP sheet developed around the opening, as shown in Figures 30 and 32. Thus, FE models were able to predict the failure sequence of the real squat SW.

## 6. Theoretical Calculation (ECP-208-2005) and (ACI 440.2R-17)

The design of shear walls with openings is stated in a very few provisions. For instance, the Euro-code (2004a) [62] stated that the design of shear walls could be done using the variable inclination truss model, the column design method, or the strut-and-tie modeling, and the Architectural Institute of Japan (AIJ) [63] determined that the response of the shear wall depends on the size, shape, and location of openings and limited the strength reduction factor of a shear wall due to the openings to 0.6 by restricting the maximum ratio of opening dimensions to the corresponding wall dimensions, while the seismic code of China GB50011-2010 [64] provided that the limiting value of the opening should be 15% of the area of wall.

According to ACI-318-19 [1] section (18.10.8), as shown in Figure 33, openings cause the formation of vertical wall segments known as wall piers and horizontal wall segments known as horizontal wall piers. The vertical wall piers are sensitive to shear collapse during earthquakes, and these elements of the walls act as columns. As a result, the ACI-318 states that the design of these parts must meet the exact requirements for seismic column design.

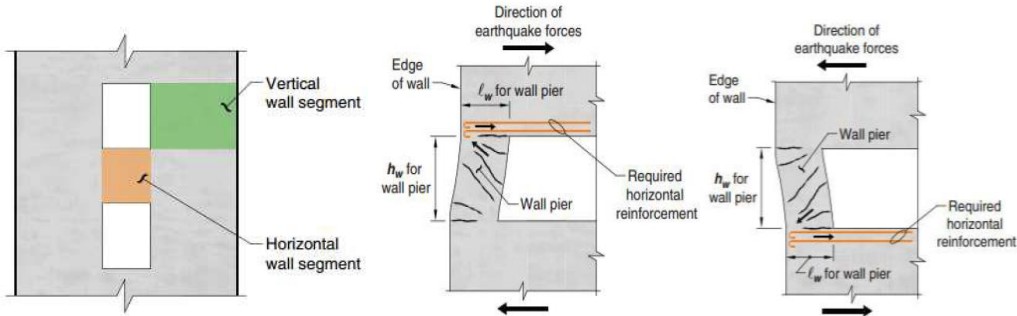

**Figure 33.** Wall with opening and required horizontal reinforcement in wall segments above and below wall piers at the edge of a wall reproduced from ref [1].

The next section will present the theoretical calculations of the shear strength capacity of the tested squat shear walls in accordance with ECP and ACI codes. Table 14 illustrates the values of cracking strength and working strength before applying the GFRP sheet and ultimate shear strength after strengthening.

**Table 14.** Shear strength capacity of squat SW according to ECP and ACI codes.

| Specimen Name | ECP-203-2020 ECP-208-2005 | | | | | | ACI-318 2019 ACI-440-2R-17 | | | | |
| --- | --- | --- | --- | --- | --- | --- | --- | --- | --- | --- | --- |
| | Before Strengthening | | | After Strengthening | | | Before Strengthening | | | After Strengthening | |
| | $A_{cv}$ mm² | $A_{cv}$ kN | $V_w$ kN | $A_f$ mm² | $V_f/\gamma_f$ kN | $V_u$ kN | $\frac{P}{f'_c A_g}$ | $V_{cr}$ kN | $V_n$ kN | $\Psi_f V_f$ kN | $V_u$ kN |
| Control | 52,500 | 57.44 | 173.3 | —— | —— | —— | 0.04 | 66.02 | 171.2 | —— | —— |
| (SW–L/4–B) | 39,380 | 43.08 | 130.0 | 94.50 | 18.14 | 148.1 | 0.04 | 52.02 | 128.4 | 25.90 | 154.2 |

**Table 14.** *Cont.*

| Specimen Name | ECP-203-2020 ECP-208-2005 | | | | | | ACI-318 2019 ACI-440-2R-17 | | | | |
| | Before Strengthening | | | After Strengthening | | | Before Strengthening | | | After Strengthening | |
| | $A_{cv}$ mm$^2$ | $A_{cv}$ kN | $V_w$ kN | $A_f$ mm$^2$ | $V_f/\gamma_f$ kN | $V_u$ kN | $\frac{P}{f_c' A_g}$ | $V_{cr}$ kN | $V_n$ kN | $\Psi_f V_f$ kN | $V_u$ kN |
|---|---|---|---|---|---|---|---|---|---|---|---|
| (SW–L/3–B) | 35,000 | 38.29 | 115.5 | 126.0 | 24.19 | 139.7 | 0.04 | 47.35 | 114.1 | 34.50 | 148.6 |
| (SW–L/2–B) | 26,250 | 28.72 | 86.60 | 189.0 | 36.29 | 122.9 | 0.04 | 38.01 | 85.60 | 51.70 | 137.3 |
| (SW–L/4–M) | 39,380 | 43.08 | 130.0 | 126.0 | 24.19 | 154.2 | 0.04 | 52.02 | 128.4 | 34.50 | 162.9 |
| (SW–L/3–M) | 35,000 | 38.29 | 115.5 | 168.0 | 32.26 | 147.8 | 0.04 | 47.35 | 114.1 | 46.00 | 160.1 |
| (SW–L/2–M) | 26,250 | 28.72 | 86.60 | 252.0 | 48.38 | 135.0 | 0.04 | 38.01 | 85.60 | 68.90 | 154.5 |
| (SW–L/4–T) | 39,380 | 43.08 | 130.0 | 94.50 | 18.14 | 148.1 | 0.04 | 52.02 | 128.4 | 25.90 | 154.2 |
| (SW–L/3–T) | 35,000 | 38.29 | 115.5 | 126.0 | 24.19 | 139.7 | 0.04 | 47.35 | 114.1 | 34.50 | 148.6 |
| (SW–L/2–T) | 26,250 | 28.72 | 86.60 | 189.0 | 36.29 | 122.9 | 0.04 | 38.01 | 85.60 | 51.70 | 137.3 |

*6.1. Shear Strength Capacity:*

For ECP-208 code [13], the nominal shear strength of an FRP-strengthened concrete member shall be determined by adding the nominal shear strength of the FRP to the nominal shear strengths of the concrete and the reinforcing steel, as given in the ECP-203-2020 as follows:

$$q_{\text{total}} = q_u + q_f \tag{9}$$

$$q_u = \left( 0.9\alpha_c\sqrt{\frac{f_{cu}}{\gamma_c}} + \rho_l\frac{f_{yl}}{\gamma_s} \right) = \frac{Q_u}{l_w * t_w} \tag{10}$$

$$\alpha_c = 0.25 \quad \text{For} \quad h_w/l_w \leq 1.5 \text{ and } \rho_l = \frac{A_l}{t_w \cdot s} \tag{11}$$

$$q_{cu} = 0.9\alpha_c\delta\sqrt{\frac{f_{cu}}{\gamma_c}} = \frac{Q_{cr}}{l_w * t_w} \text{ MPa} \tag{12}$$

$$q_{u.\max} = 0.7\sqrt{\frac{f_{cu}}{\gamma_c}} \leq 4.4 \text{ MPa} \tag{13}$$

In the case of applying axial compression load, the concrete shear strength capacity is multiplied by a factor $\delta$, which equals $\left( 1 + 0.07\left(\frac{P_u}{A_c}\right) \leq 1.5 \right)$, and the nominal shear strength of the FRP shear reinforcement was calculated as follows:

$$q_{fu} = A_f(E_f\varepsilon_{ef}/\gamma_f)(\sin\alpha + \cos\alpha)(d_f/d)/(s_f \cdot b_w) \tag{14}$$

$$A_f = 2nt_f w_f \tag{15}$$

$$E_f = f_{fu} \cdot \varepsilon_{fu} \tag{16}$$

$$\varepsilon_{fu}^* = CE \cdot \varepsilon_{fu} \text{ For (Glass/Epoxy)} \rightarrow CE = 0.75 \tag{17}$$

$$\varepsilon_{ef} = 0.75 * \varepsilon_{fu}^* \leq 0.004 \tag{18}$$

For ACI 440-2R-17 code [11], the shear strength of an FRP-strengthened concrete member is calculated using the next equations; the term $V_n$ includes the concrete and internal reinforcing steel contributions to shear capacity, and the term $A_{cv}$ refers to concrete area resisting shear:

$$V_{cr} = 0.27\sqrt{f_c'}t_w d + \frac{Nd}{4l_w} \tag{19}$$

$$V_n = V_n^* + \Psi_f V_f \tag{20}$$

$$\text{For two-side retrofit}: \ V_f = 2 \cdot t_f \cdot \varepsilon_{ef} \cdot E_f \cdot d_{fv} \tag{21}$$

$$V_n = \left( \alpha_c \lambda \sqrt{f'_c} + \frac{P_u}{6A_g} + \rho_l f_{yl} \right) A_{cv} \leq 0.66 \sqrt{f'_c} A_{cv} \tag{22}$$

$$\alpha_c = 0.25 \text{ for } h_w / l_w \leq 1.5, \text{ and } \Psi_f = 0.95 \tag{23}$$

$$A_{cv} = t_w \cdot l_w \ mm^2 \tag{24}$$

Generally, according to ECP and ACI codes, as shown in Table 14, before strengthening, the shear capacity of all shear walls was almost the same, whereas the added value of the shear strength due to FRP from the ACI code was greater than that from ECP code. The comparison between the experimental results and the ECP and ACI results is shown in Table 15.

**Table 15.** Comparison between experimental results, ECP results, and ACI results.

| Specimen Name | ECP and Exp. | | | | | | ACI and Exp. | | | | | |
|---|---|---|---|---|---|---|---|---|---|---|---|---|
| | Before Strengthening | | | After Strengthening | | | Before Strengthening | | | After Strengthening | | |
| | $P_w$ kN (Exp.) | $V_w$ kN (ECP) | $V_w/P_w$ | $P_F$ kN (Exp.) | $V_u$ kN (ECP) | $V_u/P_F$ | $P_w$ kN (Exp.) | $V_w$ kN (ACI) | $V_w/P_w$ | $P_F$ kN (Exp.) | $V_u$ kN (ACI) | $V_u/P_F$ |
| Control | 190.0 | 173.3 | 0.91 | —— | —— | —— | 190.0 | 171.2 | 0.90 | —— | —— | —— |
| (SW–L/4–B) | 142.8 | 130.0 | 0.91 | 172.27 | 148.1 | 0.86 | 142.8 | 128.4 | 0.90 | 172.27 | 154.2 | 0.90 |
| (SW–L/3–B) | 90.99 | 115.5 | 1.27 | 148.71 | 139.7 | 0.94 | 90.99 | 114.1 | 1.25 | 148.71 | 148.6 | 0.99 |
| (SW–L/2–B) | 61.15 | 86.60 | 1.41 | 86.84 | 122.9 | 1.41 | 61.15 | 85.60 | 1.38 | 86.84 | 137.3 | 1.57 |
| (SW–L/4–M) | 101.05 | 130.0 | 1.27 | —— | 154.2 | —— | 101.05 | 128.4 | 1.26 | —— | 162.9 | —— |
| (SW–L/3–M) | 80.00 | 115.5 | 1.44 | 147.88 | 147.8 | 0.99 | 80.00 | 114.1 | 1.42 | 147.88 | 160.1 | 1.08 |
| (SW–L/2–M) | 56.06 | 86.60 | 1.54 | 91.76 | 135.0 | 1.46 | 56.06 | 85.60 | 1.52 | 91.76 | 154.5 | 1.67 |
| (SW–L/4–T) | 118.63 | 130.0 | 1.09 | 170.73 | 148.1 | 0.87 | 118.63 | 128.4 | 1.08 | 170.73 | 154.2 | 0.91 |
| (SW–L/3–T) | 106.32 | 115.5 | 1.09 | 160.60 | 139.7 | 0.87 | 106.32 | 114.1 | 1.07 | 160.60 | 148.6 | 0.93 |
| (SW–L/2–T) | 54.00 | 86.60 | 1.60 | 105.81 | 122.9 | 1.16 | 54.00 | 85.60 | 1.57 | 105.81 | 137.3 | 1.30 |

Turning to details, after strengthening, for the 6.25% opening, the ECP and ACI predicted shear strength capacity results were smaller than the experimental results by a ratio ranging between 11.70–16.44% at the three studied positions. For the 11.11% opening, the ECP- and ACI-predicted shear strength capacity results were almost similar to the experimental ones. On the other hand, the predicted results from ECP and ACI for the 25% opening were greater than the experimental results by a ratio of 17.14–67.40% for all three opening positions. It is also worth mentioning that for the three different opening sizes (6.25, 11.11, and 25%), the effect of opening location was not considered in the ACI and ECP equations. As a result, there are differences between the shear capacity of the experimental results and ACI and ECP equations, but these differences are reasonable and within range. However, with the larger opening size of 25%, the results diverge as the wall behavior changes to the frame action, as mentioned before.

### 6.2. Displacement Capacity

There are limited equations to predict the displacement capacity of squat shear wall, and it is still necessary to develop better expressions to calculate the drift capacity of squat SW. Table 16 and Figure 34 show some of the proposed equations and their corresponding tri-linear backbone models. In this research, the equations proposed by Hidalgo et al. (2000) [65] were not used, as the effect of stiffness degradation due to opening was not considered, while the equations developed by Carrillo (2010) [66], Sánchez (2013) [67], and ASCE 41-13 [68,69] were used to calculate the lateral displacement, the model provided by ASCE 41-13 proposed fixed values of wall height for the estimation of the drift ratio capacity, but in this research, the used value of wall height owing to the cut-off opening was taken equal to $(h_w - h_o)$, where ($h_o$ is the opening hight), and for the same reason, the

contribution of the horizontal reinforcement ($\Delta_{\rho h}$) to calculate the ultimate displacement according to Sánchez (2013) was ignored.

**Table 16.** Proposed equations for calculating displacement capacity of squat SW.

| Proposed Models | $V_{cr}$ kN and $\Delta_{cr}$ mm | $V_{peak}$ kN and $\Delta_{peak}$ mm | $V_u$ kN and $\Delta_u$ mm |
|---|---|---|---|
| Hidalgo et al. (2000) [65] | $V_{cr} = \alpha\sqrt{f'_c}A_{cw}$ <br> $\Delta_{cr} = \left(0.00175\frac{M}{Vl_w}\right)h_w$ <br> $for\ \frac{M}{Vl_w} \leq 1$ | $V_n = (\alpha_c\sqrt{f'_c} + \rho_n f_{yl})A_{cv}$ <br> $\Delta_{peak} = \left(0.0027 + 0.0033\frac{M}{Vl_w}\right)h_w$ <br> $for\ \frac{M}{Vl_w} \leq 1$ | $V_u = 0.8V_{peak}$ <br> $\Delta_u = \left(0.002 + 0.012\frac{M}{Vl_w}\right)h_w$ <br> $for\ \frac{M}{Vl_w} \leq 1$ |
| Carrillo (2010) [66] | $V_{cr} = \alpha\sqrt{f'_c}A_{cw}$ <br> $R_{cr}\% = \left(\frac{V_{cr}}{K_{cr}}\right)\frac{100}{h_w}$ <br> $K_{cr} = 1/\left(\frac{h_w^3}{c_1 3E_c I_g} + \frac{h_w}{c_2 G_c A_{cw}}\right)$ <br> $(c_1 = c_2 = 1)$ stiffness modifier | $V_n = (\alpha_c\sqrt{f'_c} + \rho_n f_{yl})A_{cv}$ <br> $R_{peak}\% = \frac{1}{5200}\frac{V_{peak}}{t_w\sqrt{f'_c}}e^{1.3\frac{M}{Vl_w}}$ <br> For deformed bar | $V_u = 0.8V_{peak}$ <br> $R_u\% = \frac{1}{3650}\frac{(0.8*V_{peak})}{t_w\sqrt{f'_c}}e^{1.35\frac{M}{Vl_w}}$ <br> For deformed bar |
| Sánchez (2013) [67] | $V_{cr} = \left(0.32 - 0.045\frac{M}{Vl_w}\right)\sqrt{f'_c}A_{cw}$ <br> $\Delta_{cr-f} = \left(0.01 + 0.005\frac{h_w}{l_w}\right)\frac{h_w}{100}$ <br> $\Delta_{cr} = \left(0.025 + 0.13\frac{M}{Vl_w}\right)\frac{h_w}{100}$ | $V_n = (\alpha_c\sqrt{f'_c} + \rho_n f_{yl})A_{cv}$ <br> $\Delta_{peak} = \Delta_{flexural} + \Delta_{shear}$ <br> $\Delta_{peak} = \frac{V_{peak}\ h_w^3}{3E_c(0.7I_g)} + \frac{V_{peak}\ l_w}{A_w\sqrt{f'_c}}\frac{1}{300}e^{1.33\frac{M}{Vl_w}}$ <br> $E_c = 4700\sqrt{f'_c}MPa$ | $V_u = 0.8V_{peak}$ <br> $\Delta_u = (\Delta_{peak} + \Delta_{\rho h}) \times (0.6\frac{M}{Vl_w} + 0.5)$ <br> $\Delta_u \geq 1.2\Delta_{peak}$ <br> $\Delta_{\rho h} = 9/\rho_h f_{yh} \leq 9mm$ |
| ASCE 41-13 (Wallace 2007) [68,69] | $V_{cr} = 0.33\sqrt{f'_c}\sqrt{1 + \frac{P/A_g f'_c}{48.2\sqrt{f'_c}}}A_w$ <br> $V_{cr} \leq 0.6V_{peak}$ <br> $\Delta_{cr} = \Delta_{f.cr} + \Delta_{s.\ cr}$ <br> $\Delta_{cr} = \frac{V_{cr}h_w^3}{3c_1 E_c I_g} + \frac{V_{cr}h_w}{c_2 G_c A_w}$ <br> $G_c = \frac{E_c}{2(1+v)} \cong 0.42E_c$ | $V_n = (\alpha_c\sqrt{f'_c} + \rho_n f_{yl})A_{cv}$ <br> $\frac{\Delta_{peak}}{h_w} = 1\%$ for $P/t_w l_w f'_c \leq 0.05$ | $V_u = 0.2V_{peak}$ <br> $\frac{\Delta_u}{h_w} = 2\%$ for $P/t_w l_w f'_c \leq 0.05$ |

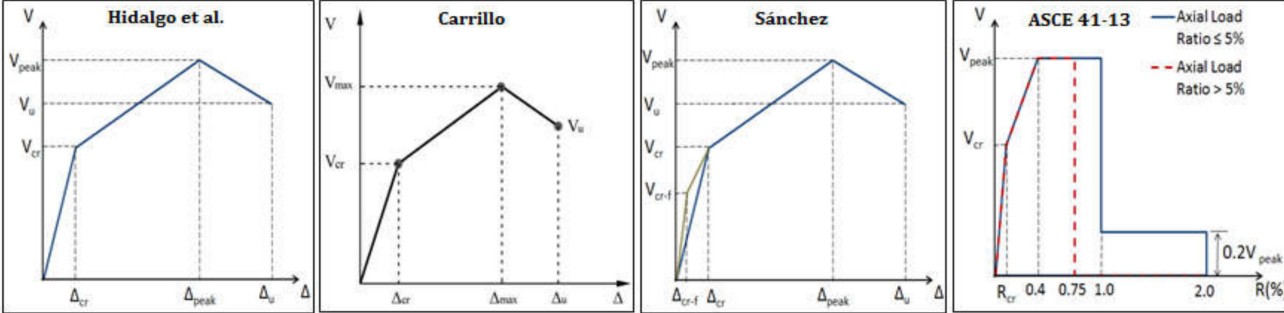

**Figure 34.** Tri-linear backbone models reproduced from [65–69].

From Table 17, it is worth mentioning that the effect of opening position and applying strengthening on the displacement capacity was not considered in the proposed equations except the results from FE analysis. By comparing the lateral displacement capacity of the aforementioned proposed models to the tested values, it was found that:

- The results from nonlinear FE models are the most relevant results to experimental results, as the effect of cut-out openings and their position were considered and succeeded by the results from the ASCE 41-13 model and Sánchez (2013) model.
- The results from the Carrillo (2010) model do not seem to agree with the experimental observations; the reason is that the stiffness degradation due to opening size was not considered.
- By using finite-element models, a faster, accurate, and more economical design could be made, as the FE analysis overcomes the inaccuracy and difficulty of theoretical nonlinear analysis [70–72].

**Table 17.** Comparing lateral displacement capacity to experimental results.

| Specimen Name | Carrillo (2010) [66] | | | Sánchez (2013) [67] | | | ASCE 41-13 [68,69] | | | ANSYS 14.5 [52] | | |
|---|---|---|---|---|---|---|---|---|---|---|---|---|
| | $\Delta_{peak}$ mm | $\Delta_u$ mm | $\frac{\Delta_w}{\Delta_{peak}}$ | $\Delta_{peak}$ mm | $\Delta_u$ mm | $\frac{\Delta_w}{\Delta_{peak}}$ | $\Delta_{peak}$ mm | $\Delta_u$ mm | $\frac{\Delta_w}{\Delta_{peak}}$ | $\Delta_{peak}$ mm | $\Delta_u$ mm | $\frac{\Delta_w}{\Delta_{peak}}$ |
| Control | 2.62 | 3.14 | 2.89 | 6.84 | 8.21 | 1.12 | 7.50 | 15.00 | 1.02 | 8.56 | —— | 0.89 |
| (SW–L/4–B) | 1.97 | 3.82 | 5.84 | 5.75 | 8.28 | 2.01 | 5.63 | 11.26 | 2.06 | 8.21 | 9.12 | 1.41 |
| (SW–L/3–B) | 1.75 | 3.68 | 2.55 | 5.51 | 8.61 | 0.81 | 5.00 | 10.00 | 0.90 | 3.52 | 5.72 | 1.28 |
| (SW–L/2–B) | 1.31 | 3.40 | 2.93 | 5.53 | 10.63 | 0.70 | 3.75 | 7.50 | 1.03 | 3.13 | 5.52 | 1.24 |
| (SW–L/4–M) | 1.97 | 4.03 | 2.27 | 5.75 | 8.75 | 0.78 | 5.63 | 11.26 | 0.80 | 3.00 | 4.32 | 1.50 |
| (SW–L/3–M) | 1.75 | 3.96 | 5.09 | 5.51 | 9.28 | 1.63 | 5.00 | 10.00 | 1.79 | 5.81 | 6.62 | 1.54 |
| (SW–L/2–M) | 1.31 | 3.82 | 2.76 | 5.53 | 11.97 | 0.66 | 3.75 | 7.50 | 0.97 | 3.20 | 5.31 | 1.14 |
| (SW–L/4–T) | 1.97 | 3.82 | 1.56 | 5.75 | 8.28 | 0.54 | 5.63 | 11.26 | 0.55 | 2.54 | 4.58 | 1.22 |
| (SW–L/3–T) | 1.75 | 3.68 | 4.36 | 5.51 | 8.61 | 1.39 | 5.00 | 10.00 | 1.53 | 5.64 | 6.96 | 1.36 |
| (SW–L/2–T) | 1.31 | 3.40 | 8.15 | 5.53 | 10.63 | 1.95 | 3.75 | 7.50 | 2.87 | 3.11 | 5.14 | 3.46 |

## 7. Proposed Shear Strength's Reduction Factors (β)

For similar cases of squat shear walls with various size openings at different positions, our experimental and analytical findings suggest the following reduction factors (β) for the shear strength capacity equation (Equation (10)) provided by ECP-203-2020:

- For shear walls with a 6.25% opening, the proposed reduction factor is about (0.55) for the middle opening and (0.65–0.75) for top and bottom openings.

$$V_M = (0.55)V_{control} \tag{25}$$

$$V_T = (0.65)V_{Control} \tag{26}$$

$$V_B = (0.75)V_{Control} \tag{27}$$

- For shear walls with an 11.11% opening, the proposed reduction factor is about (0.45) for the middle opening and (0.55) for top and bottom openings.

$$V_M = (0.45)V_{control} \tag{28}$$

$$V_T = V_B = (0.55)V_{Control} \tag{29}$$

- For shear walls with a 25% opening, the proposed reduction factor is about (0.3) for the middle and top openings and (0.35) for bottom openings.

$$V_M = V_T = (0.3)V_{Control} \tag{30}$$

$$V_B = (0.35)V_{Control} \tag{31}$$

From Table 18, the mean ratios of the experimental to the estimated shear strength ($V_{proposed}/V_{Exp.}$) are close to (1.00). Finally, the author's point of view recommends further experimental and analytical research to gain a better understanding of the behavior of squat shear walls with openings and emphasize the accuracy of these findings.

**Table 18.** Comparison between the Estimated Equation and Experimental Results.

| Specimen Name | $P_w$ kN (Exp.) | $V_w$ kN (ECP) | Proposed reduction Factor β | $V_{proposed} = \beta V_w$ kN | $\frac{V_{propsed}}{P_{w\ (Exp.)}}$ |
|---|---|---|---|---|---|
| Control | 190.0 | 173.3 | —— | 173 | —— |
| (SW–L/4–B) | 142.8 | 130.0 | 0.75 | 129.8 | 0.99 |
| (SW–L/3–B) | 90.99 | 115.5 | 0.55 | 95.15 | 1.05 |
| (SW–L/2–B) | 61.15 | 86.60 | 0.35 | 60.55 | 0.99 |

**Table 18.** *Cont.*

| Specimen Name | $P_w$ kN (Exp.) | $V_w$ kN (ECP) | Proposed reduction Factor β | $V_{proposed} = \beta V_w$ kN | $\frac{V_{propsed}}{P_{w\ (Exp.)}}$ |
|---|---|---|---|---|---|
| (SW–L/4–M) | 101.05 | 130.0 | 0.55 | 95.15 | 0.94 |
| (SW–L/3–M) | 80.00 | 115.5 | 0.45 | 77.85 | 0.97 |
| (SW–L/2–M) | 56.06 | 86.60 | 0.30 | 51.9 | 0.93 |
| (SW–L/4–T) | 118.63 | 130.0 | 0.65 | 112.5 | 0.95 |
| (SW–L/3–T) | 106.32 | 115.5 | 0.55 | 95.15 | 0.90 |
| (SW–L/2–T) | 54.00 | 86.60 | 0.30 | 51.9 | 0.96 |

## 8. Conclusions and Recommendations:

In this paper, an experimental study, nonlinear finite-element simulation, and comparison between different code provisions of squat shear walls with openings were conducted, and the assessment of using GFRPs sheets for strengthening around the post-cut-out opening was undertaken. Based on this study, the following conclusions have been made:

1.  The control squat SW behaves as a shear-controlled member, and the load-deflection curve did not show a clear yielding point, especially after installing a cut-out opening; further, the squat SW with a bigger opening exhibited the highest loss in stiffness and lateral load capacity as compared to others, and these walls start to behave like a frame action.
2.  For the walls with the same opening dimension but with different locations, it was observed that the middle opening position resulted in the highest loss in lateral load capacity compared to other opening positions
3.  The strengthening scheme using the GFRP sheet around opening resulted in increasing lateral load capacity by a value ranging from (42.01–95.94%) and enhancing displacement capacities, so the author suggests that in the case of inserting a cut-out opening in the squat walls, it is necessary to apply FRP material around these openings to achieve a safe response and restore their integrity and serviceability.
4.  Both the experimental and the finite-element (FE) analysis results were comparable.
5.  The shear capacity prediction made by ACI and ECP was comparable to the experimental results up to an 11.11% opening, but with the greater opening size, the results were not reliable.
6.  There are several theoretical models developed by various researchers for the analysis of shear wall strength and displacement, but their results were not accurate due to the difficulty of nonlinear analysis.
7.  After analyzing the tri-linear backbone curves, it is clear that some of the offered models, such as ASCE 41-13 and that of Sánchez, were able to anticipate displacement capacity; however, the Carrillo model was underestimated.
8.  Further experimental and analytical researches are required to emphasize the accuracy of the proposed reduction factors for the shear strength capacity equation provided by ECP code.
9.  It is recommended for future work to investigate different shapes of openings, different types of FRPs based on capacity and cost, increase the number of applied layers with various schemes, and study more parameters including concrete and steel strength as well as the effect of adding different types of fiber to concrete mix [73–78].

**Author Contributions:** Conceptualization, H.M. and N.Z.; Data curation, H.M. and A.F.D.; Formal analysis, N.Z. and M.H.; Funding acquisition, A.F.D.; Methodology, M.H. and A.S.; Supervision, A.S.; Writing—original draft, M.H.; Writing—review & editing, A.F.D. All authors have read and agreed to the published version of the manuscript.

**Funding:** This research received no external funding.

**Institutional Review Board Statement:** Not applicable.

**Informed Consent Statement:** Not applicable.

**Data Availability Statement:** All data are included in the manuscript.

**Conflicts of Interest:** The authors declare no conflict of interest.

**Notations**

| Symbol | Description | f |
|---|---|---|
| $A_f$ | Area of FRP external reinforcement | $mm^2$ |
| $A_g$ | Gross area of the wall | $mm^2$ |
| $b_c$ | The width of the concrete | Mm |
| $b_f$ | The width of the FRP sheet | Mm |
| C1 | Factor for flexural stiffness reduction due to cracking | |
| C2 | Factor for reduced shear stiffness due to cracking | |
| CE | Environmental reduction factor | |
| d | Effective depth of the concrete section | mm |
| $d_f$ | Depth of FRP shear reinforcement but not to exceed $h_w$ | mm |
| $d_{fv}$ | The effective depth of shear wall | mm |
| $E_c$ | Modulus of elasticity of concrete | $N/mm^2$ |
| $E_f$ | Modulus of elasticity of FRP | $N/mm^2$ |
| $E_x, E_y, E_z$ | Elastic moduli of FRP composites in $x$, $y$, and $z$ directions | $N/mm^2$ |
| f | Stress at any strain $\varepsilon$ | $N/mm^2$ |
| $f_{cu}$ | Cube compressive strength of concrete | $N/mm^2$ |
| $f'_c$ | Cylinder compressive strength of concrete $f'_c = 0.8 f_{cu}$ | $N/m^2$ |
| $f_f$ | Tensile strength of the FRP | $N/m^2$ |
| $f_t$ | Maximum tensile strength of concrete | $N/mm^2$ |
| $f_y$ | Yield stress of steel | $N/mm^2$ |
| $G_f$ | Fracture energy of concrete | $N/m^2 \times m$ |
| $G_{xy}, G_{xz}, G_{yz}$ | Shear modulus of FRP composites for the $xy$, $xz$, and $yz$ planes | $N/m^2 \times m$ |
| H | Height of the wall | mm |
| $H_0$ | Height of the opening in the wall | mm |
| L | Length of the wall | Mm |
| $\left(\frac{M}{Vl_w}\right)$ | Wall shear span-to-length ratio | |
| $q_u$ | Nominal shear strength | $N/mm^2$ |
| $q_{cu}$ | Nominal shear strength of concrete | $N/mm^2$ |
| $q_{su}$ | Nominal shear strength of reinforcing steel | $N/mm^2$ |
| $q_{fu}$ | Nominal shear strength of FRP | $N/mm^2$ |
| $t_f$ | Nominal thickness of one ply of the FRP reinforcement | Mm |
| $V_n$ | Nominal shear strength | $N/mm^2$ |
| $V_f$ | The shear strength provided by the FRP. | $N/mm^2$ |
| $V_n^*$ | The nominal shear strength of the existing shear wall | $N/mm^2$ |
| $w_f$ | Width of the FRP reinforcing plies | Mm |
| $\beta_c$ | Shear transfer coefficients for a closed crack | |
| $\beta_o$ | Shear transfer coefficients for an open crack | |
| $\beta_w$ | Width ratio factor | |
| $\varepsilon$ | Strain at stress f | mm/mm |
| $\varepsilon_o$ | Strain at ultimate compressive strength $f'_c$ | mm/mm |
| $\varepsilon_{ef}$ | Effective strain in FRP reinforcement | mm/mm |
| $\varepsilon^*_{fu}$ | Maximum strain in FRP | mm/mm |
| $\tau$ | Local bond stress | $N/mm^2$ |
| $\tau_{max}$ | Maximum local bond stress | $N/mm^2$ |
| S | Local slip | mm |
| $S_0$ | Local slip at maximum local bond stress | mm |
| $\upsilon$ | Poisson's ratios | |
| $\upsilon_{xy}, \upsilon_{xz}, \upsilon_{yz}$ | Major Poisson's ratios of FRP composites for the xy, xz, and yz planes | |
| $\rho_l$ | Ratio of area of distributed longitudinal reinforcement to gross concrete area perpendicular to that reinforcement | |
| $\rho_t$ | Ratio of area of distributed transverse reinforcement to gross concrete area perpendicular to that reinforcement | |
| $\Phi$ | Strength reduction factor | |
| $\gamma_c$ | Material strength reduction factor of concrete | |
| $\gamma_s$ | Material strength reduction factor of steel reinforcement. | |
| $\gamma_f$ | Material strength reduction factor of FRP shear reinforcement | |
| $\alpha_{;f}$ | Angle of inclination of FRP reinforcement to the longitudinal axis of the member | |
| $\alpha_c$ | Coefficient defining the relative contribution of the concrete to shear strength | |
| $_f\Psi$ | FRP strength reduction factor = 0.95 for shear fully wrapped section | |

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
