# Peer review of "Performance of Strengthened, Reinforced Concrete Shear Walls with Opening"

_sustainability, doi:10.3390/su142114366_

Round 1

Reviewer 1 Report

This article carries out experimental studies and numerical simulations using the finite element method of strength, stiffness and deformation of Strengthened Reinforced Concrete shear walls with different sizes and hole placements.

The experimental and numerical results performed by the authors were compared with Egyptian and North American technical standards, and also with results from three other articles.

The work follows the appropriate scientific methodology with citation of a vast and representative bibliography on the subject.

English writing could be improved.

From the analysis of the results obtained, it can be noted that:

(i) there are finite element results that are insecure relative to the experimental ones. The authors explain the reasons in the conclusions, indicating that in finite element analysis a better characterization of the behavior of materials is required (also taking cracking into account), which they end up not doing in the article.

(ii) numerical and finite element results indicate influence of hole positioning on the results, which is not specified in the standards. This seems to be the most important result of the work since most of the other results were already expected (loss of performance with the introduction of holes, influence of the dimensions of the holes, increase in performance with the addition of FRP).

(iii) the authors do not present formulation proposals to take into account the influence of hole positioning, nor do they suggest future works. This would constitute, in fact, an innovation on this subject that already has several works published.

Finally, one wonders how this subject, the use of reinforcements with FRP in shear walls, can fit into a publication on sustainability.

The article can be published after these comments have been answered and the corrections indicated in the attached file have been made.

Author Response

Dear editor,

 Thank you for the valuable comments, appreciated it. Please see attached.

Reviewer 2 Report

This paper presents an experimental and analytical study on the behavior of reinforced concrete (RC) shear walls with openings strengthened using glass-fiber-reinforced-polymer (GFRP) sheets. The structure of the article is complete, the content is detailed and innovative, but the article still has some minor problems:

1, the table and picture layout is too chaotic not beautiful enough

2, the introduction to cite some more references in the last year or two.

3, the conclusion is simplified to highlight the key points

In summary, the article is accepted after further revision.

Author Response

Thank you for your valuable comments, appreciated it. Please see attached,

Reviewer 3 Report

Please further elaborate on the novelty of your work in abstract.

Please further describe the main steps that you followed and the main outstanding outcomes in the abstract.

The presented introduction is pretty modest. Please include a brief but critical review regarding the conducted research studies in the introduction.  It is recommended to add a section “research significance” and highlight the main contribution of your findings.

 Please include the latest research studies related to your work preferably between 2019 and 2022. Accordingly, please further explain the performance of walls (for example masonry walls) with openings by including a summary of work titled Evaluating the behaviour of centrally perforated unreinforced masonry walls: Applications of numerical analysis, machine learning, and stochastic methods.

Moreover, please consider the effect of GFRP sheets on enhancing the performance of strengthened columns by considering the article titled Effect of Fiber Reinforced Polymer Tubes Filled with Recycled Materials and Concrete on Structural Capacity of Pile Foundations.

Please further elaborate on the sample preparation process, the limitations and the main assumptions on the test procedures.

Please explain the verification procedure of the conducted experiments vs the FEM analyses..

Please state more about the boundary condition of the performed analyses and outliers based on the performed analyses.

Please include a comparative discussion based on the reported results in Figures 25 and 26.

Please include a compelling discussion on the main parameters that can affect the results of this work.

Please revise the conclusion and offer a quantitative-based approach on reporting the main outcomes of this study.

Author Response

(The authors gave the same response as above.)

Round 2

Reviewer 3 Report

N/A